# Reduction of Derlin activity suppresses Notch-dependent tumours in the *C. elegans* germ line

**Ramya Singh**[¤a¤b], **Ryan B. Smit, Xin Wang, Chris Wang**[¤c]**, Hilary Racher**[¤d¤e]**, Dave Hansen***

Department of Biological Sciences, University of Calgary, Calgary, Canada

¤a Current address: Department of Biology, McGill University, Montréal, Canada
¤b Current address: Institute for Research in Immunology and Cancer (IRIC), Université de Montréal, Montréal, Canada
¤c Current address: Department of Sciences, Ambrose University, Calgary, Canada
¤d Current address: Dynacare, 115 Midair Court, Brampton, Canada
¤e Current address: Department of Laboratory Medicine & Pathobiology, University of Toronto, Toronto, Canada
* dhansen@ucalgary.ca

**Data Availability Statement:** All relevant data are within the manuscript and its Supporting Information files.

## Abstract

Regulating the balance between self-renewal (proliferation) and differentiation is key to the long-term functioning of all stem cell pools. In the *Caenorhabditis elegans* germline, the primary signal controlling this balance is the conserved Notch signaling pathway. Gain-of-function mutations in the GLP-1/Notch receptor cause increased stem cell self-renewal, resulting in a tumour of proliferating germline stem cells. Notch gain-of-function mutations activate the receptor, even in the presence of little or no ligand, and have been associated with many human diseases, including cancers. We demonstrate that reduction in CUP-2 and DER-2 function, which are Derlin family proteins that function in endoplasmic reticulum-associated degradation (ERAD), suppresses the *C. elegans* germline over-proliferation phenotype associated with *glp-1(gain-of-function)* mutations. We further demonstrate that their reduction does not suppress other mutations that cause over-proliferation, suggesting that over-proliferation suppression due to loss of Derlin activity is specific to *glp-1/Notch (gain-of-function)* mutations. Reduction of CUP-2 Derlin activity reduces the expression of a read-out of GLP-1/Notch signaling, suggesting that the suppression of over-proliferation in Derlin loss-of-function mutants is due to a reduction in the activity of the mutated GLP-1/Notch (GF) receptor. Over-proliferation suppression in *cup-2* mutants is only seen when the Unfolded Protein Response (UPR) is functioning properly, suggesting that the suppression, and reduction in GLP-1/Notch signaling levels, observed in Derlin mutants may be the result of activation of the UPR. Chemically inducing ER stress also suppress *glp-1(gf)* over-proliferation but not other mutations that cause over-proliferation. Therefore, ER stress and activation of the UPR may help correct for increased GLP-1/Notch signaling levels, and associated over-proliferation, in the *C. elegans* germline.

**Funding:** This work was supported by the Natural Sciences Research Council of Canada (06647-2015) and Canadian Institute of Health Research (PJT-155999) to DH. The funders had no role in study design, data collection and analysis, decision to publish, or preparation of the manuscript. https://www.nserc-crsng.gc.ca/index_eng.asp https://cihr-irsc.gc.ca/e/193.html.

**Competing interests:** The authors have declared that no competing interests exist

## Author summary

Notch signaling is a highly conserved signaling pathway that is utilized in many cell fate decisions in many organisms. In the *C. elegans* germline, Notch signaling is the primary signal that regulates the balance between stem cell proliferation and differentiation. Notch gain-of-function mutations cause the receptor to be active, even when a signal that is normally needed to activate the receptor is absent. In the germline of *C. elegans*, gain-of-function mutations in GLP-1, a Notch receptor, results in over-proliferation of the stem cells and tumour formation. Here we demonstrate that a reduction or loss of Derlin activity, which is a conserved family of proteins involved in endoplasmic reticulum-associated degradation (ERAD), suppresses over-proliferation due to GLP-1/Notch gain-of-function mutations. Furthermore, we demonstrate that a surveillance mechanism utilized in cells to monitor and react to proteins that are not folded properly (Unfolded Protein Response-UPR) must be functioning well in order for the loss of Derlin activity to supress over-proliferation caused by *glp-1/Notch* gain-of-function mutations. This suggests that activation of the UPR may be the mechanism at work for suppressing this type of over-proliferation, when Derlin activity is reduced. Therefore, decreasing Derlin activity may be a means of reducing the impact of phenotypes and diseases due to certain Notch gain-of-function mutations.

## Introduction

Stem cell populations provide the source material for future tissue generation and play an important role in the development and maintenance of many tissues. A defining feature of stem cells, their ability to both self-renew and differentiate, is key to their function. Stem cells must maintain a balance between self-renewal and differentiation as excessive self-renewal can lead to tumour formation while too much differentiation leads to a depleted stem cell pool. The decision to self-renew or differentiate is essential for the proper development of their tissues. Critical systems like this require many layers of redundancy in order to have a high level of robustness[1,2]. This way, if pressure is applied to one layer, other layers are able to ensure proper decision-making. These layers of redundancy can also allow external inputs to impinge on the system, giving it the ability to adapt. Understanding these layers of redundancy will aid in research in stem cell-related diseases and in using stem cells as therapeutic agents.

An indirect mechanism to modulate stem cell systems is through the regulation of protein folding and protein quality control. For example, recently it has been shown that increasing the genesis of misfolded proteins in hematopoietic stem cells (HSCs) impairs self-renewal of HSCs [3]. In the case of muscle stem cells, impairment of autophagy, the lysosomal degradation of long-lived proteins and damaged organelles, leads to senescence and stem cell exhaustion [4]. As another example, the transcription factor NRF3 is significantly mutated across twelve cancer cell lines and promotes cancer cell proliferation [5,6]. NRF3 is regulated by ER retention and endoplasmic-reticulum-associated degradation (ERAD), two cellular mechanisms responsible for surveillance of protein folding [7,8]. ERAD is a multi-step process in which misfolded proteins are recognized, retrotranslocated into the cytoplasm and targeted for degradation by the proteasome [8–11]. Recognition of misfolded proteins involves lectins and chaperone proteins in the ER [8]. Retrotranslocation occurs in protein complexes containing E3 ubiquitin ligases that also ubiquitinate the misfolded protein. In yeast, the Doa10/Ubc7 complex retrotranslocates and ubiquitinates proteins with misfolded cytosolic domains (ERAD-C pathway), while the Hrd1/Hrd3/Der1 complex acts on proteins with misfolded ER luminal domains (ERAD-L pathway) [12,13]. This distinction between different ERAD

pathways is less clear in mammalian systems [9,14]. In both yeast and mammals, a p97 (Cdc48)/Npl4/Ufd1 complex extracts many of the targeted proteins in an ATPase-dependent manner allowing them to be degraded by the proteasome [12,15].

The housekeeping function of ERAD has important physiological implications for protein homeostasis. For example, as much as 75% of the wild type cystic fibrosis transmembrane conductance regulator (CFTR) protein is targeted for degradation through ERAD [16–20]. Single amino acid mutations in the 140 kDa, twelve transmembrane domain CFTR protein disrupt its proper folding such that all CFTR protein is degraded by ERAD leading to cystic fibrosis [17][21]. Dysregulation of ERAD can lead to the accumulation of misfolded proteins in the ER, which induces ER stress [14]. In response to ER stress, a series of protective cellular events are triggered to deal with the accumulation of misfolded proteins. Translation is attenuated to limit protein folding burden, the ER expands and becomes more elaborate to increase protein folding capacity, expression of chaperone proteins is increased and expression of ERAD components are increased to assist in protein folding. Collectively this response is called the Unfolded Protein Response (UPR)[14,22]. If the UPR fails, apoptosis can be triggered to eliminate the stressed cell in both animal models and human disease[14,23]. The increased levels of protein synthesis required for overproliferation in cancer cells is thought to increase basal levels of ER stress and the UPR [24]. This increase in ER stress is thought to either make cancer cells more resilient, or more susceptible to artificially inducing ER stress [25,26]. Understanding how cancer cells (and all stem cells) regulate and respond to ER stress is crucial in order to be able to understand how therapeutics act on them [27].

The adult *C. elegans* germline harbours stem cells whose self-renewal is regulated by the GLP-1/Notch signalling pathway. The gonads of the *C. elegans* hermaphrodite comprise of two U-shaped tubes that meet at a common uterus [28,29]. Germ cells are born at the distal end of each arm and mature as they move proximally towards the vulva. The most distal population of germ cells are mitotically dividing stem cells [29]. As cells move proximally, they enter into meiosis and mature in an assembly line like manner to produce gametes. The pool of distal stem cells is maintained by GLP-1/Notch signalling and loss of GLP-1/Notch signalling results in a loss of the stem cell population, while increased GLP-1/Notch signalling leads to tumour formation [30–32] (S1A and S1B Fig). Two redundant pathways comprising of GLD-1 and GLD-2 function downstream of GLP-1/Notch signaling to promote differentiation. If the activities of both these pathways is reduced or eliminated a germline tumour results, similar to that due to increased GLP-1/Notch signaling [33–35] (S1C Fig). Many other regulatory controls ranging from factors controlling cell division such as CYE-1/CDK-2, subunits of the DNA polymerase alpha-primase complex, or proteasomal activity to signalling pathways such as MPK-1 ERK, Insulin, TGF- β, and TOR have been identified that modulate the balance between self-renewal vs differentiation to provide robust control [36–42]. They also add a layer of modulatory control necessary for the germline to adapt. Disruption of any one of these modulatory controls have weak effects on the balance between self-renewal and differentiation in an otherwise wildtype genetic background under ideal conditions; however, the redundancy of these pathways combine to create a robust system necessary for the balance between germline stem cell self-renewal and differentiation.

CUP-2 is a member of the Derlin (degradation in the ER) family of proteins, that function in ERAD [14,43–45]. Derlins were initially discovered in yeast with Der1 and Dfm1 [45,46]. Mammals have three Derlin family members, Derlin-1, Derlin-2 and Derlin-3 [47]. CUP-2 is most similar to human Derlin-1 and a second *C. elegans* Derlin, DER-2, is most similar to human Derlin-2 and Derlin-3 [43,47,48]. As expected for proteins functioning in ERAD, loss of *cup-2* and *der-2* result in activation of the UPR [43,44]. Further support for *cup-2* and *der-2*'s role in ERAD is provided by the fact that as is the case with yeast *der1*, *cup-2* is also

synthetically lethal with *ire-1*, a sensor for the UPR and overexpression of *der-2* in Δ*der1* Δ*ire1* yeast strains partially suppresses the conditional lethality associated with the strain [43,46,49].

CUP-2 (coelomocyte uptake defective) was first identified in *C. elegans* as being required for endocytosis by the scavenger-like cells, the coelomocytes [50]. Later, CUP-2 was found to bind SNX-1 (sorting nexin), a component of the retromer complex in early endosomes [51,52]. This interaction was also observed with human Derlins and Sorting Nexins [52]. Similar to its role in ERAD, in endocytosis CUP-2 is thought to aid in recognition of misfolded plasma membrane proteins and their transportation to the ER for degradation [52].

As part of another study to identify potential mRNA targets of PUF-8 (to be published elsewhere), we found that loss of *cup-2* activity strongly suppressed the overproliferation phenotype observed in *puf-8(0); glp-1(gf)* animals (see below). PUF-8 is a pumilio homolog that is known to play a role in the proliferation vs differentiation balance in the *C. elegans* germline and loss of *puf-8* strongly enhances the stem cell overproliferation phenotype of *glp-1(gf)* mutants [53–55]. Although *cup-2* does not appear to be a direct target of PUF-8 (S2 Fig), the loss of *cup-2* suppressing overproliferation provides an inroad into studying the role of ERAD in affecting GLP-1/Notch signaling and the proliferation vs. differentiation decision. Previous studies have shown that worms mutant for the ERAD component CUP-2, have increased expression of HSP-4::GFP, a hallmark of ER stress, as well as activation of the Unfolded Protein Response (UPR) [43,44]. These studies have also shown that *der-2*, the other worm Derlin, is partially redundant with *cup-2* in the activation of the UPR. Here we investigate the effect that these Derlin mutants and ER stress have on the balance between stem cell proliferation and differentiation in the *C. elegans* germ line.

Recently, studies have highlighted an emerging link between Notch signalling, ER stress and the UPR. Disruptions to ER zinc homeostasis affect Notch trafficking and activity in human cancer cell lines and *Drosophila* imaginal wing discs [56,57]. Mutations in p97, a key ERAD component, also disrupt Notch signalling in *Drosophila* wing development [58]. Inducing ER stress in human cell culture induces expression of the Notch ligand DLL4 [59]. Whether ERAD plays a role in physiological Notch signalling, or whether Notch signalling is modulated only in response to ER stress is unclear from these latest studies.

Here we describe the effect that ERAD and the UPR have on the Notch-dependent control of stem cell proliferation in the *C. elegans* germline. We show that germline tumour formation resulting from increased GLP-1/Notch signalling is suppressed by mutations in *cup-2* and *der-2*, encoding Derlin proteins which are components of ERAD. We also show that ectopically induced ER stress suppresses germline stem cell over-proliferation caused by increased GLP-1/Notch signaling and that this suppression requires the UPR. Both loss of *cup-2* and induced ER stress can only suppress GLP-1/Notch-dependent tumours, suggesting they act directly on the Notch pathway, in a context-specific manner that suppresses excess overproliferation. We propose that ER stress and the UPR have a protective role, correcting for aberrant over-proliferation caused by increased GLP-1/Notch signaling levels in the *C. elegans* germline and restoring proper balance between stem cell proliferation and differentiation. Therefore, this study contributes to our understanding of how affecting protein folding capacity by modulating ER stress, can regulate the balance between stem cell self-renewal and differentiation.

## Results

### Loss of *cup-2* suppress *glp-1(gf)* tumours in correlation with the strength of GLP-1/Notch signalling

We analyzed *cup-2*'s relationship with key *glp-1* alleles (S1 Table). As mentioned above, our initial observation leading us to investigate the role of ERAD in the proliferation vs.

differentiation decision was the partial suppression of overproliferation in *puf-8(0); glp-1(gf)* animals through the loss of *cup-2* activity. As we have previously reported, loss of *puf-8* strongly enhances the overproliferation phenotype of *glp-1(gf)* alleles [54]. To distinguish proliferating stem cells from differentiating cells, we analyzed germline phenotypes using antibodies against REC-8 (marker for proliferating cells in mitosis) and HIM-3 (marker for differentiating cells entering meiosis)[35,60,61] (Fig 1A). We found that while 95% of *puf-8 (q725); glp-1(oz264gf)* gonads were completely tumourous, with no evidence of cells entering meiosis as measured by α-REC-8 and α-HIM-3 staining, only 14% of gonads were completely tumourous when *cup-2* activity was also removed (Fig 1B and Table 1). Indeed, while no phenotypically wild-type *puf-8(q725); glp-1(oz264gf)* gonads were observed, 22% of *cup-2 (tm2838); puf-8(q725); glp-1(oz264)* were wild-type with no evidence of overproliferation. Therefore, the loss of *cup-2* activity strongly suppresses the overproliferation observed in *puf-8 (q725); glp-1(oz264gf)* animals.

We first more fully characterized the suppression of *puf-8(0); glp-1(gf)* and asked whether the suppression by loss of *cup-2* activity correlated with the strength of the *glp-1* gain-of-function allele used. Loss of *puf-8* in *glp-1(gf)* mutants results in overproliferation, with the degree of overproliferation appearing to be dependent on the strength of the *glp-1* gain-of-function allele [54]. We analyzed three different *glp-1(gf)* alleles, *ar202*, *oz264* and *ar224*, which in a *puf-8(0)* mutant background results in 100%, 95% and 86% gonads being fully tumourous at 20°C, respectively (Fig 1B and Table 1). When *cup-2* activity was also removed in these backgrounds the percentage of fully tumourous gonads was reduced to 92%, 14% and 41%, respectively. Therefore, the ability of *cup-2(0)* to suppress overproliferation is not *glp-1(gf)* allele-specific, and the degree of suppression is at least somewhat correlated with the relative strength of the *glp-1(gf)* allele.

To determine if the suppression by *cup-2(0)* is dependent on loss of *puf-8* activity, we tested whether *cup-2(0)* could suppress other overproliferation mutants that are due to increased GLP-1/Notch signaling, but wild-type for *puf-8*. We found that *cup-2(0)* partially suppresses overproliferation observed in an *rfp-1(ok572); glp-1(oz264gf)* double mutant background. RFP-1 is an E3 ligase that promotes proteasomal degradation of proliferation promoting proteins in order to allow germ cells to differentiate [62]. In *rfp-1(ok572); glp-1(oz264gf)* 61% of gonads contain an incomplete tumour, while 39% of the gonads appear wild-type for proliferation (Fig 1B and Table 1). Loss of *cup-2* strongly suppresses this overproliferation phenotype, with 93% of the gonads appearing wild-type and only 7% having an incomplete tumour when *cup-2* activity is also removed (Fig 1B and Table 1). *cup-2(0)* also suppresses overproliferation in *glp-1(gf)* single mutants. For this analysis we counted the number of cells in the distal proliferative zone at 20°C (Fig 2A). Wild-type gonads contain ~221 cells in this region, while *glp-1 (ar202gf)* and *glp-1(oz264gf)* gonads contain ~610 and ~384 cells respectively (Fig 2A and 2B and Table 2), demonstrating significant overproliferation. The *cup-2(tm2838)* single mutant contains ~193 cells, slightly fewer than in wild-type gonads (Table 2). Removal of *cup-2* strongly suppressed *glp-1(ar202gf)* and *glp-1(oz264gf)* overproliferation, with only ~195 and ~226 cells in the respective double mutants (Table 2). We noticed that although *glp-1(ar224gf)* single mutants do not show an overtly large proliferative zone size at 20°C, removal of *cup-2*, was also able to slightly suppress the proliferative zone size in this genetic background. Therefore, *cup-2(0)'s* suppression of germline overproliferation is not dependent on the loss of *puf-8*.

We have demonstrated that loss of *cup-2* activity suppresses *glp-1(gf)* mediated overproliferation. We reasoned that if *cup-2* interacts with the GLP-1/Notch signaling pathway then its loss may also enhance the temperature sensitive partial loss-of-function *glp-1(bn18)* allele [63]. We found that at 20°C, *cup-2(0); glp-1(bn18)* gonads have ~95 cells in the proliferative zone, while *cup-2(0)* and *glp-1(bn18)* have ~195 and ~138, respectively (Fig 2B) (Table 2). While it does appear that *cup-2(0)* enhances the smaller proliferative zone phenotype of *glp-1(bn18)*, we

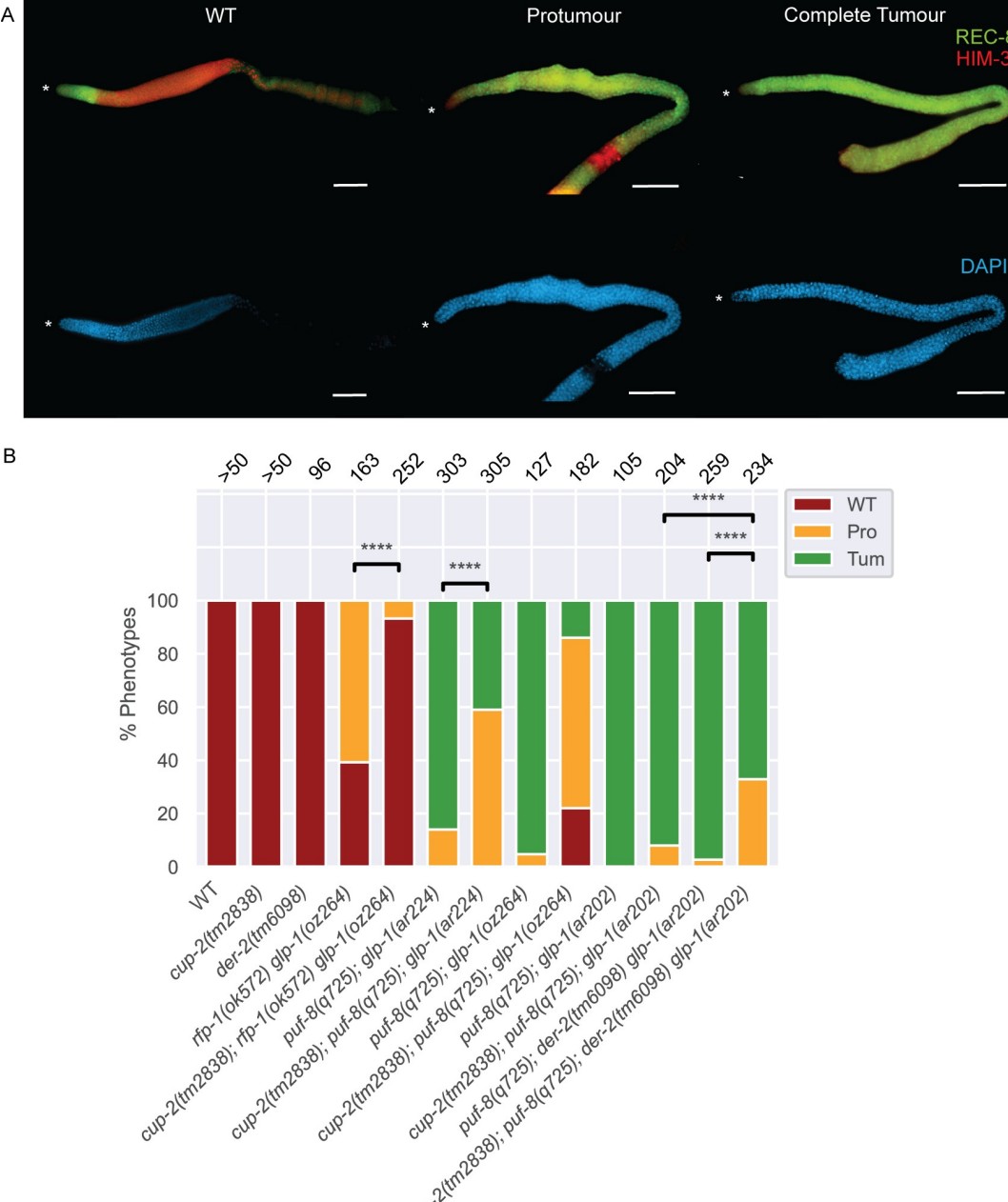

**Fig 1. Derlin mutants suppress Notch-dependent tumours in correlation with the strength of GLP-1/Notch signalling. A.** Representative images of wild-type (WT), protumourous (Pro) and completely tumourous (Tum) phenotypes as visualized by α-REC-8(green) and α-HIM-3(red) immunostaining. DAPI staining was used to visualize nuclei. Wild-type (WT) is defined as a gonad arm with distal α-REC-8(+) cells followed by α-HIM-3(+) cells and presence of both sperm and oocytes in the proximal arm of the gonad. A protumour is defined as a gonad arm containing both α-REC-8(+) and α-HIM-3(+) cells, but not differentiated sperm and oocytes. Proximal tumourous gonad arms with only sperm but no oocytes were counted as protumours. Gonad arms with mostly only α-REC-8(+) and a few α-HIM-3(+) positive cells were also counted as protumourous. Complete tumour is defined as a gonad arm that contains only α-REC-8(+) cells and no α-HIM-3(+) cells. Both tumourous phenotypes are of the *cup-2(tm2838); puf-8(q725); der-2(tm6098) glp-1(ar202)* genetic background. Asterisk, distal tip. Scale bar = 20μm. **B.** Quantification of phenotypic analysis of the dissected gonads of the indicated genotypes analyzed by α-REC-8(green) and α-HIM-3(red) immunostaining. In this and all other bar graphs, numbers on the top of the bars indicate the number of gonads analyzed. Chi-square test; **** = p ≤ 0.0001.

**Table 1. Loss of Derlin activity supresses *glp-1(gf)* overproliferation.**

| Genotype | WT[1] | Protumour[2] | Complete Tumour[3] | n[4] |
|---|---|---|---|---|
| Wild-type[5] | 100% | 0% | 0% | > 50 |
| *cup-2(0)*[6] | 100% | 0% | 0% | > 50 |
| *der-2(0)*[7] | 100% | 0% | 0% | 96 |
| *rfp-1(0)*[8] *glp-1(oz264gf)* | 39% | 61% | 0% | 163 |
| *cup-2(0); rfp-1(0) glp-1(oz264gf)* | 93% | 7% | 0% | 252 |
| *puf-8(0)*[9]*; glp-1(ar224gf)* | 0% | 14% | 86% | 303 |
| *cup-2(0); puf-8(0); glp-1(ar224gf)* | 0% | 59% | 41% | 305 |
| *puf-8(0); glp-1(oz264gf)* | 0% | 5% | 95% | 127 |
| *cup-2(0); puf-8(0); glp-1(oz264gf)* | 22% | 64% | 14% | 182 |
| *puf-8(0); glp-1(ar202gf)* | 0% | 0% | 100% | 105 |
| *cup-2(0); puf-8(0); glp-1(ar202gf)* | 0% | 8% | 92% | 204 |
| *puf-8(0); der-2(0) glp-1(ar202gf)* | 0% | 3% | 97% | 259 |
| *cup-2(0); puf-8(0); der-2(0) glp-1(ar202gf)* | 0% | 33% | 67% | 234 |

[1]Wild-type (WT) is defined as a gonad arm with distal α-REC-8(+) cells followed by α-HIM-3(+) cells and presence of both sperm and oocytes in the proximal arm of the gonad

[2]A protumour is defined as a gonad arm containing both α-REC-8(+) and α-HIM-3(+) cells, but not differentiated sperm and oocytes. Proximal tumourous gonad arms with only sperm but no oocytes were counted as protumours. Gonad arms with mostly only α-REC-8(+) and a few α-HIM-3(+) positive cells were also counted as protumourous.

[3]Complete tumour is defined as a gonad arm that contains only α-REC-8(+) cells and no α-HIM-3(+) cells

[4]Number of gonad arms

[5]N2

[6]*cup-2(tm2838)*

[7]*der-2(tm6098)*

[8]*rfp-1(ok572)*

[9]*puf-8(q725)*

do not consider this to be a strong enhancement because enhancement is not to the point of causing a Glp (germ line proliferation defective) phenotype in which no proliferative cells are present [30], which is observed with other enhancers of *glp-1(bn18)* [64]. Even at 22.5°C we do not observe enhancement resulting in Glp animals (S2 Table). This suggests that loss of *cup-2* has a stronger effect on gain-of-function alleles of *glp-1* than loss-of-function alleles.

## Loss of the *cup-2* paralog, *der-2*, also suppress *glp-1(gf)* mediated overproliferation

*cup-2* encodes a Derlin protein that has previously been shown to be involved in ERAD [43,44], functioning partially redundantly with DER-2, the other *C. elegans* Derlin protein (Schaheen et al. 2009). DER-2 is thought to be the functional ortholog of yeast Der1p since overexpression of *C. elegans* DER-2 in yeast *Δder1 Δire1* strains partially restores degradation of an Der1p associated ERAD substrate and partially suppresses the conditional lethality phenotype of the double mutant [46]. If disruption of ERAD is responsible for *cup-2(0)*'s ability to suppress germline overproliferation, then we would expect loss of *der-2* to likewise suppress overproliferation. We found that loss of *der-2* does decrease the size of the distal proliferative zone in *glp-1(ar202gf)* animals from ~610 cells in *glp-1(ar202gf)* to ~405 in *der-2(tm6098) glp-1(ar202gf)* double mutants (Fig 2A and 2C and Table 2). Importantly, the size of the distal proliferative zone in *der-2(tm6098)* single mutants is similar to that in wild-type animals (~227 and ~221 respectively), suggesting that the suppression of *glp-1(ar202gf)* is not simply due to

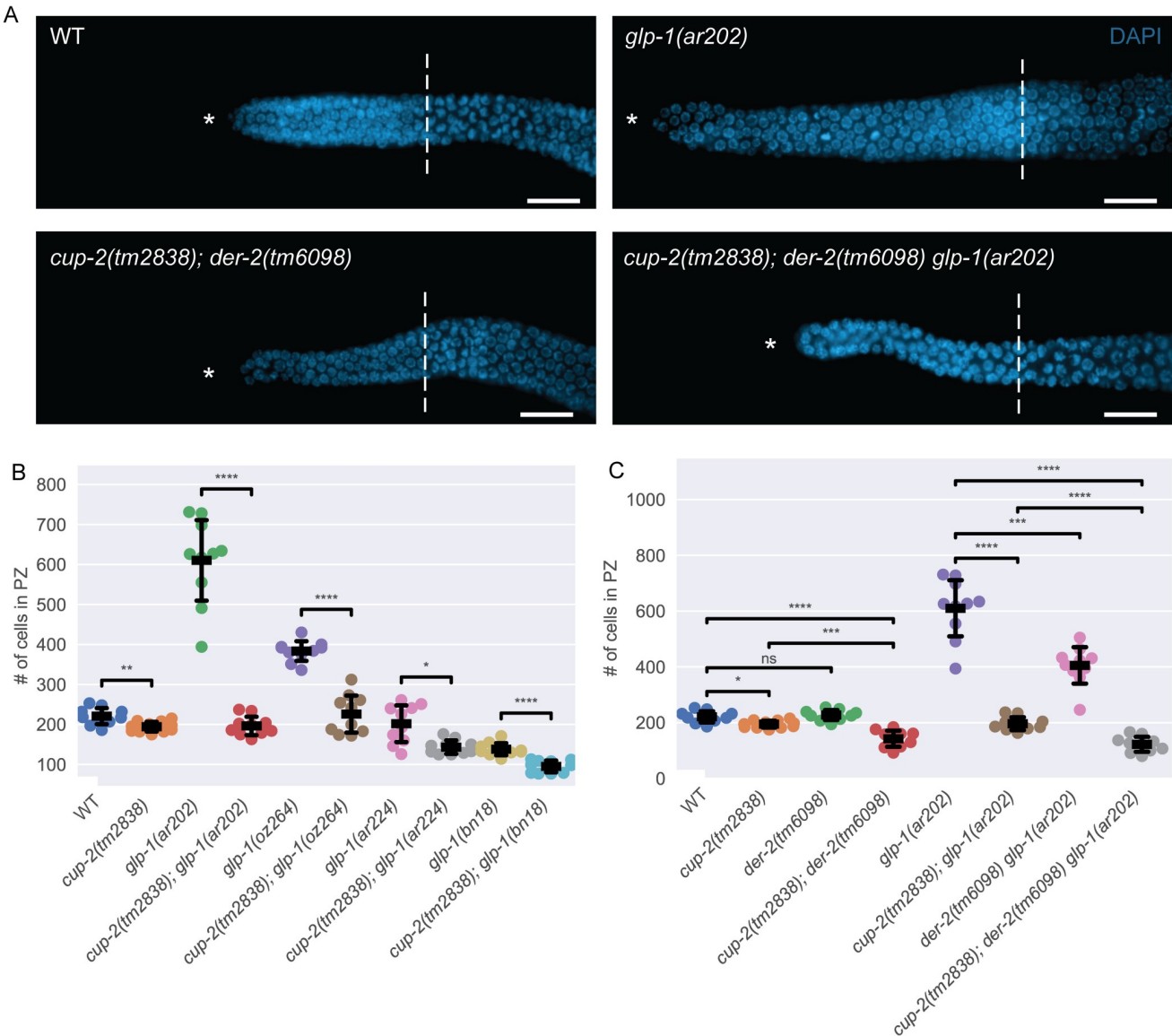

**Fig 2. Derlin mutants have smaller proliferative zone sizes.** A. Representative images of distal germlines of wildtype(WT), *cup-2(tm2838); der-2 (tm6098)*, *glp-1(ar202)* and *cup-2(tm2838); der-2(tm6098) glp-1(ar202)* worms stained by DAPI to visualize nuclei. Asterisk, distal tip. Dashed line, transition zone. Scale bar = 10μm. B. *cup-2(tm2838)* suppresses the proliferative zone sizes of *glp-1(ar202)*, *glp-1(oz264)*, *glp-1(bn18)* and germlines. Total number of cells in the proliferative zone of the indicated genotypes. C. Derlin mutants additively suppress the size of the proliferative zone of WT and *glp-1 (ar202)* germlines. Total number of cells in the proliferative zone of the indicated genotypes. In this and subsequent scatter plots, thick line = mean; error bars = standard deviation; t-test independent samples with Bonferroni correction; ns = 0.05 < p ≤ 1; * = 0.01 < p ≤ 0.05, ** = 0.001 < p ≤ 0.01, *** = 0.0001 < p ≤ 0.001, **** = p ≤ 0.0001.

an overall reduced rate of proliferation. The suppression of *glp-1(ar202gf)* is most pronounced when the activities of both *cup-2* and *der-2* are removed (Fig 2A and 2C and Table 2), reducing the size of the proliferative zone from ~610 to ~122. This suggests that *cup-2* and *der-2* may have some redundant function. We also noticed that genotypes mutant for both *cup-2* and *der-2* tend to have slightly narrower and smaller gonads, overall (Fig 2A). Indeed, the size of the distal proliferative zone in the *cup-2(tm2838); der-2(tm6098)* double mutant (~142) is smaller than either single mutant (~195 for *cup-2* and ~227 for *der-2*), or the wild-type proliferative zone (~221)(Fig 2C and Table 2). Moreover, while loss of *cup-2* suppresses the complete

**Table 2. Loss of Derlin activity suppress *glp-1(gf)* overproliferation in the proliferative zone.**

| Genotype | Average no. of cells in the proliferative zone[10] | S.D. | min | max | n[11] |
|---|---|---|---|---|---|
| Wild-type[12] | 221 | 21 | 186 | 253 | 10 |
| *cup-2(0)*[13] | 195 | 13 | 175 | 215 | 12 |
| *der-2(0)*[14] | 227 | 19 | 194 | 255 | 10 |
| *cup-2(0); der-2(0)* | 142 | 30 | 92 | 184 | 10 |
| *glp-1(ar202gf)* | 610 | 106 | 394 | 731 | 10 |
| *cup-2(0); glp-1(ar202gf)* | 196 | 24 | 163 | 237 | 10 |
| *der-2(0) glp-1(ar202gf)* | 405 | 69 | 246 | 505 | 10 |
| *cup-2(0); der-2(0) glp-1(ar202gf)* | 122 | 28 | 80 | 166 | 10 |
| *glp-1(bn18ts)* | 138 | 16 | 114 | 171 | 10 |
| *cup-2(0); glp-1(bn18ts)* | 95 | 16 | 77 | 113 | 10 |
| *glp-1(ar224gf)* | 202 | 48 | 126 | 261 | 10 |
| *cup-2(0); glp-1(ar224gf)* | 143 | 18 | 125 | 176 | 10 |
| *glp-1(oz264gf)* | 384 | 26 | 336 | 430 | 10 |
| *cup-2(0); glp-1(oz264gf)* | 226 | 49 | 172 | 312 | 10 |

[10]Proliferative cells defined as cells distal to the transition zone as identified by crescent-shaped nuclei stained by DAPI

[11]Number of gonad arms for which the total number of cells in the proliferative zone were counted

[12]N2

[13]*cup-2(tm2838)*

[14]*der-2(tm6098)*

tumour phenotype from 100% tumourous gonads in *puf-8(q725); glp-1(ar202gf)* double mutants to 92% tumourous animals in *cup-2(tm2838); puf-8(q725); glp-1(ar202)* triple mutants, also eliminating *der-2* function in *cup-2(tm2838); puf-8(q725); der-2(tm6098) glp-1 (ar202)* quadruple mutants significantly reduces the percentage of completely tumourous animals to 67% (Fig 1B and Table 1). Therefore, the suppression of *glp-1(ar202gf)* (proliferative zone counts), and the suppression of *puf-8(0); glp-1(ar202gf)* by loss of *cup-2* and *der-2* suggests that *cup-2* and *der-2* function redundantly in promoting robust germline proliferation.

## Derlin loss-of-function mutants have smaller proliferative zone sizes

In order to ascertain whether Derlin mutants affect cell proliferation, we first asked whether the size of the proliferative zone is altered in Derlin mutants, *cup-2* and *der-2*. As noted above, we found that *cup-2* mutant worms have a slightly smaller proliferative zone than wild-type, whereas *der-2* single mutants do not significantly alter the proliferative zone size (Fig 2C and Table 2). Moreover, *cup-2; der-2* double mutants have a statistically significantly smaller proliferative zone size than wild-type ($p = 2.182 \times 10^{-5}$, t-test independent samples with Bonferroni correction) and *cup-2* single mutants ($p = 2.014 \times 10^{-4}$, t-test independent samples with Bonferroni correction) (Fig 2C and Table 2). This implies that *cup-2* and *der-2* may have overlapping but partially redundant roles in regulating cell proliferation.

Of the two Derlins, *cup-2*'s interaction with *glp-1* gain-of-function alleles is stronger; however, *cup-2* and *der-2* additively have the strongest effect. We conclude that Derlin's interaction with the GLP-1/Notch signalling pathway could be proportional to the strength of excessive GLP-1/Notch signalling and is most pronounced with strong *glp-1* gain-of-function alleles.

## Derlin loss-of-function mutants do not suppress Notch-independent tumours

We have demonstrated that loss of both *cup-2* and *der-2* suppress germline overproliferation due to increased GLP-1/Notch signaling. This suppression could be achieved by lowering

GLP-1/Notch signaling levels; alternatively, it could be achieved by affecting signaling down-stream or parallel to GLP-1/Notch signaling, or by directly inhibiting proliferation. To differentiate between these possibilities, we tested whether Derlin mutants could suppress germline overproliferation mutants that do not increase GLP-1/Notch signalling. The GLD-1 and GLD-2 pathways function downstream of GLP-1/Notch signaling. If the activity of just one of these two pathways is eliminated germ cells proliferate and enter meiosis similar to wild-type [33,35,65]. However, if the activities of both pathways are reduced or eliminated a germline tumour results, similar to the tumour due to increased GLP-1/Notch signaling [33–35,66]. We tested overproliferation mutants that have reduced or no GLD-1 and GLD-2 pathway activities to see if loss of Derlin activity could suppress the overproliferation.

We first tested the *gld-3(0) nos-3(0)* animals, with *gld-3(0)* reducing GLD-2 pathway activity and *nos-3(0)* reducing GLD-1 pathway activity, which form a robust GLP-1/Notch signaling independent tumour [34]. We found that no *gld-3(q730) nos-3(q650)* (n = 102) or *cup-2 (tm2838); gld-3(q730) nos-3(q650)* (n = 111) gonads showed evidence of meiotic cells, as measured by the presence of REC-8(-) HIM-3(+) cells, suggesting that loss of *cup-2* does not suppress this overproliferation (Fig 3A and 3B).

Since our analyses of Notch-dependent tumours suggest that loss of *cup-2* only weakly suppresses more robust tumourous backgrounds, we were concerned that the *gld-3(0) nos-3(0)* tumour was perhaps too strong for *cup-2(0)* to suppress. Therefore, we also analyzed *gld-2(0) gld-1(0)* animals. In these double mutants all gonads are over-proliferative [33]; however, α-REC-8 and α-HIM-3 staining reveals that many cells in the distal arm of the gonad and/or around the loop region enter meiosis (REC-8(-) HIM-3(+)), while all cells in the proximal arm appear proliferative in most gonads (Fig 3C and 3D and Table 3)[35]. We reasoned that since many cells enter meiosis in *gld-2(0) gld-1(0)* animals, this genetic background would be more sensitized for suppression of overproliferation. We found that neither loss of *cup-2* nor *der-2* suppressed *gld-2(0) gld-1(0)* tumours, as measured by an expansion of REC-8(-) HIM-3(+) cells into the proximal arm (Fig 3D and Table 3). Even simultaneous loss of *cup-2* and *der-2* did not suppress overproliferation in *gld-2(0) gld-1(0)* double mutants (Fig 3C and 3D and Table 3). The morphology of (REC-8(-) HIM-3(+)) staining in the distal arms of the quadruple mutant did not appear to be different from that of *gld-2(0) gld-1(0)* double mutants (S3 Fig). Taken together, we conclude that loss of Derlin activity is unable to suppress overproliferation that is due to loss of GLD-1 and GLD-2 pathway genes. This suggests that Derlin mutants do not suppress proliferation in general, but rather, specifically suppress GLP-1/Notch signalling.

### *cup-2* loss-of-function mutant reduces the expression of SYGL-1, a readout of GLP-1/Notch signalling, in *glp-1(gf)* gonads

We have demonstrated that loss of Derlin activity suppresses overproliferation due to increased GLP-1/Notch signalling, but not overproliferation in GLD-1/GLD-2 pathway mutants. This raises the possibility that loss of Derlin activity reduces the amount of GLP-1/Notch signalling in *glp-1(gf)* mutants. *sygl-1* is a downstream transcriptional target of GLP-1/Notch signalling in the germline, functioning redundantly with *lst-1*, and is expressed in the distal proliferating cells in the germline [67–71]. To determine the effect of loss of Derlin activity on GLP-1/Notch signalling, we analyzed SYGL-1 expression in the relevant mutants (Fig 4A).

Consistent with previous reports of SYGL-1 expression levels in *glp-1(gf)* mutants, we found that the zone of SYGL-1 expression expanded proximally in *glp-1(ar202gf)* gonads as compared to wild-type (S4B Fig)[67,68]. In wild-type gonads, SYGL-1 expression peaks around five cell diameters from the distal end, then decreases gradually until plateauing

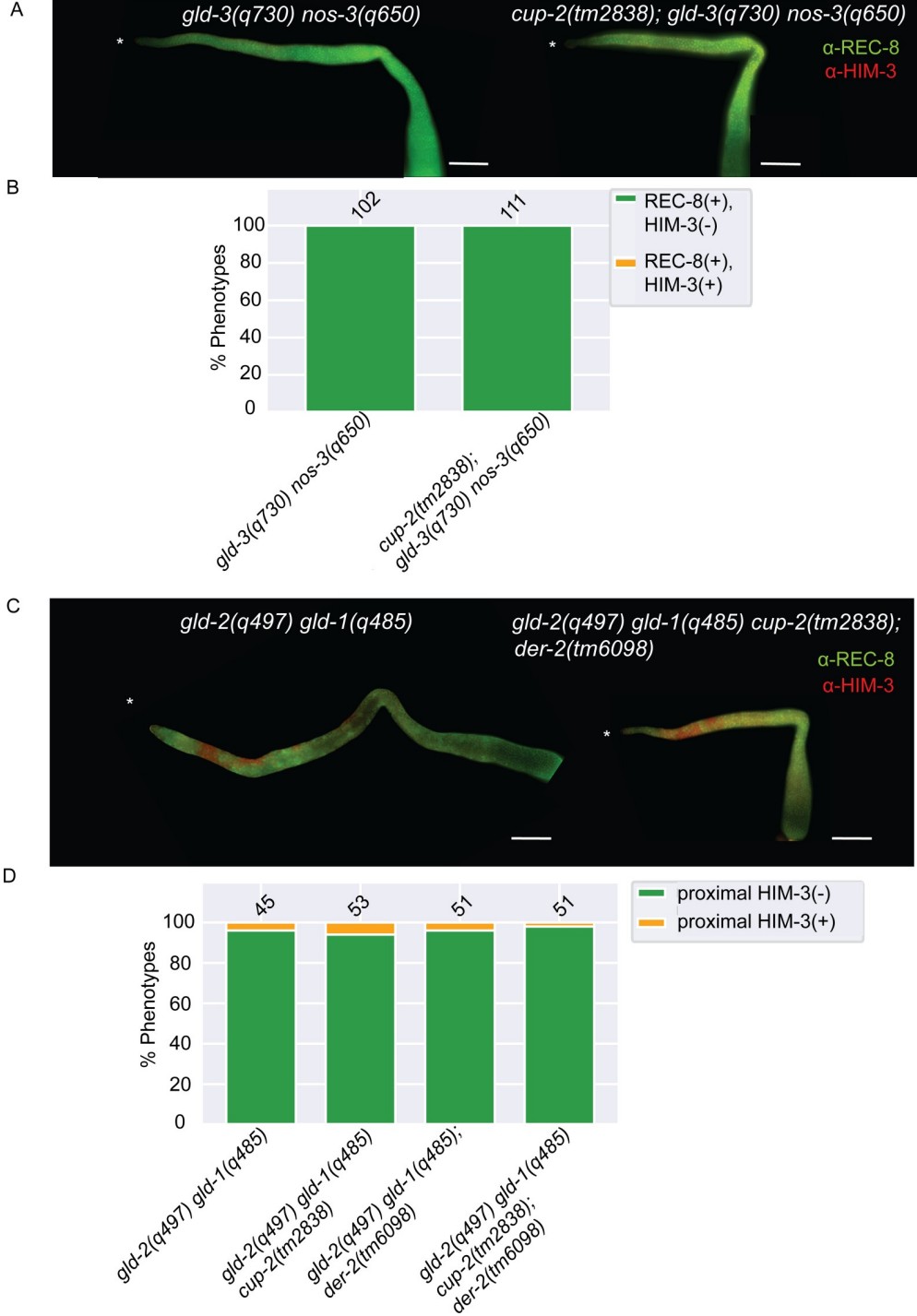

**Fig 3. Derlin mutants do not suppress Notch-independent tumours.** A. Representative images of germlines of indicated genotypes visualized by α-REC-8(green) and α-HIM-3(red) immunostaining. DAPI staining was used to visualize nuclei. No nuclei stained with meiotic entry marker HIM-3, were detected in both genotypes. Asterisk, distal tip. Scale bar = 20μm. B. Quantification of phenotypic analysis of the indicated genotypes analyzed by α-REC-8 and α-HIM-3 immunostaining. C. Representative images of germlines of indicated genotypes visualized by α-REC-8(green) and α-HIM-3(red) immunostaining. DAPI staining was used to visualize nuclei. Nuclei stained with meiotic entry marker HIM-3, were detected in in the distal arm in both genotypes but not in the proximal arm of the gonad. Asterisk, distal tip. Scale bar = 20μm. D. Quantification of phenotypic analysis of the indicated genotypes analyzed by α-REC-8 and α-HIM-3 immunostaining.

**Table 3. Loss of Derlin activity does not supress *gld-2 gld-1* Notch-independent tumours.**

| Genotype | HIM-3(-) in proximal germline | HIM-3(+) in proximal germline | n[15] |
|---|---|---|---|
| *gld-2(0) gld-1(0)*[16] | 96% | 4% | 45 |
| *gld-2(0) gld-1(0) cup-2(0)*[17] | 94% | 6% | 53 |
| *gld-2(0) gld-1(0); der-2(0)*[18] | 96% | 4% | 51 |
| *gld-2(0) gld-1(0) cup-2(0); der-2(0)*[e] | 98% | 2% | 51 |

[15]Number of gonad arms
[16]Complete genotype *gld-2(q497) gld-1(q485)*
[17]Complete genotype *gld-2(q497) gld-1(q485) cup-2(tm2838)*
[18]Complete genotype *gld-2(q497) gld-1(q485); der-2(tm6098)*
[e] Complete genotype *gld-2(q497) gld-1(q485) cup-2(tm2838); der-2(tm6098)*

around 20 cell diameters from the distal end (Fig 4B). In *glp-1(ar202gf)* gonads the peak is around seven cell diameters and the plateau is around 24 cell diameters (Fig 4B). In *cup-2 (tm2838); glp-1(ar202gf)* double mutants the expansion of SYGL-1 expression is suppressed, with the pattern of SYGL-1 expression being very similar to wild-type (Figs 4B and S4C). Interestingly, the SYGL-1 pattern in *cup-2* single mutants is also shifted ~two cell diameters distally as compared to wild-type (Figs 4B and S4A). Since the data shown in Fig 4B was generated by comparing genotypes imaged on different slides and normalizing against SYGL-1 intensity of a wild-type background as an internal control (see Materials and Methods) we wanted to more directly compare the effect of loss of *cup-2* activity on SYGL-1 expression in *glp-1(ar202gf)* animals, without normalization. Therefore, we compared SYGL-1 expression in *glp-1(ar202gf)* animals with *cup-2(tm2838); glp-1(ar202gf)* animals on the same slide (S4D Fig). This experiment yielded similar results to those obtained with normalization; the loss of *cup-2* activity supresses the expansion of SYGL-1 expression along the distal-proximal axis in *glp-1(ar202gf)* animals. The distal movement of the SYGL-1 expression pattern in *cup-2(0)* as compared to wild-type, and in *cup-2(tm2838); glp-1(ar202gf)* as compared to *glp-1(ar202gf)*, suggests that reduction of Derlin activity results in a decrease in GLP-1/Notch signaling, and that this reduction in GLP-1/Notch signaling is the likely cause of the suppression of the *glp-1(gf)* overproliferation by the loss/reduction of Derlin activity.

## CUP-2 functions in the germline to affect proliferation

Our genetic results suggest that loss of Derlin protein function, particularly CUP-2, results in reduction of excessive GLP-1/Notch signalling levels in the germline. To gain more insight into how CUP-2 may interact with GLP-1/Notch signaling we sought to determine its expression pattern. For this we tagged the endogenous *cup-2* locus with a C-terminal *v5::2xflag* epitope tag using CRISPR/Cas9 editing [72–76]. We created three independent *cup-2::v5::2Xflag* alleles (*cup-2(ug1)*, *cup-2(ug2)* and *cup-2(ug3)*), all expressing CUP-2::V5::2XFLAG as detected by an α-FLAG western blot [77]. We found that CUP-2 is expressed throughout the germline (Fig 5A). The expression levels of CUP-2 are lower in the distal end of the gonad and increase proximally. Although CUP-2 levels are low in the distal region of the gonad, where GLP-1/Notch signalling is known to be active, it is above background (Fig 5B and 5C).

Derlin proteins are known components of the ERAD protein quality control mechanism [44–47,78,79]. Therefore, if CUP-2 is involved in ERAD in the germline, we would expect to find CUP-2 localized to the endoplasmic reticulum (ER). We found that CUP-2 expression within the germline partially co-localizes with SP12, a signal peptidase that is a commonly used ER marker (Fig 5C)[80]. The expression of CUP-2 in the ER is consistent with previous

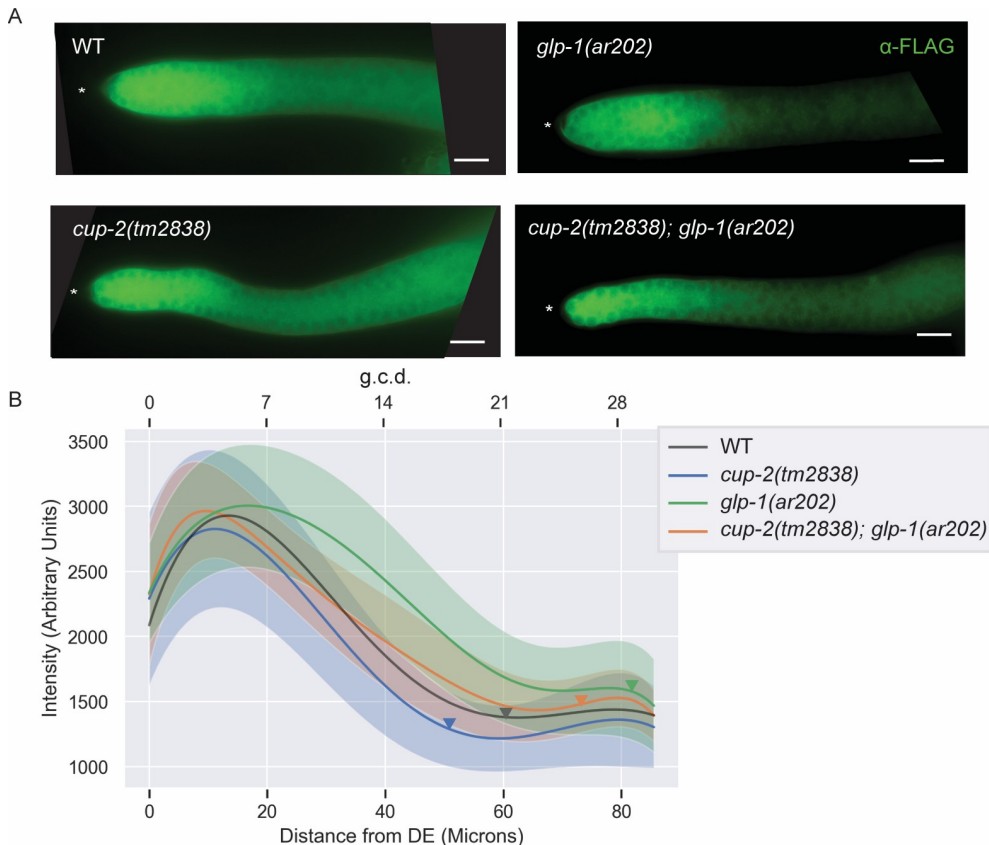

**Fig 4. Loss of *cup-2* decreases SYGL-1 protein levels in *glp-1(ar202)* germlines.** A. Representative images of SYGL-1 protein expression observed in *sygl-1(am307); glp-1(ar202)* and *cup-2(tm2838) sygl-1(am307); glp-1(ar202)* genetic backgrounds by α-FLAG immunostaining. The *sygl-1(am307)* allele represents a 3XFLAG tagged version of endogenous SYGL-1. Asterisk, distal tip. Scale bar = 10μm. B. Normalized, fitted average SYGL-1 intensities measured by α-FLAG immunostaining of the indicated genotypes, each harbouring the *sygl-1(am307)* allele. Shaded areas indicate unscaled fitted standard deviation of the intensity measurements for each genotype. Standard deviation for average *sygl-1(am307)* has not been shown for ease of visualization but can be seen in S4 Fig. Average normalized intensities and standard deviations were fit to a sixth order polynomial. Fifteen gonads were analyzed for intensity measurements. Arrowheads point to the average location of the transition zone measured in at least seven gonads of each genotype. Distances from distal end (DE) were measured in microns and converted to germ cell diameters (g.c.d) as a reference, by assuming 1 g.c.d. = 2.833 microns. WT intensity measurement shown is the average *sygl-1(am307)* intensity across the three experiments used for scaling.

studies of CUP-2 expression in coelomocytes using transgenes [43,52]. We conclude that a proportion of CUP-2 is localized to the ER in germ cells.

To better understand the mechanism by which *cup-2* participates in regulating stem cell proliferation, we asked whether *cup-2* activity is required within the germline or the somatic gonad for robust proliferation in *glp-1(gf)* animals. In *rrf-1(pk1417)* mutants RNAi is not efficiently processed in the somatic gonad but is efficiently processed in the germline [81,82]. We found that when *cup-2* was knocked down by RNAi in *rrf-1(pk1417); puf-8(q725); glp-1 (oz264gf)* animals, 63% of animals contained complete tumours as compared to 97% in the *gfp (RNAi)* negative control animals (Fig 6A and Table 4). Moreover, 16% of the gonads were wild-type (no overproliferation) as compared to no wild-type gonads in the negative control (Fig 6A and Table 4). Therefore, the overproliferation suppression by *cup-2* RNAi in a *rrf-1* mutant background suggests that *cup-2*'s role in stem cell proliferation takes place in the germline and not the somatic gonad. Although the level of suppression observed with *cup-2* RNAi

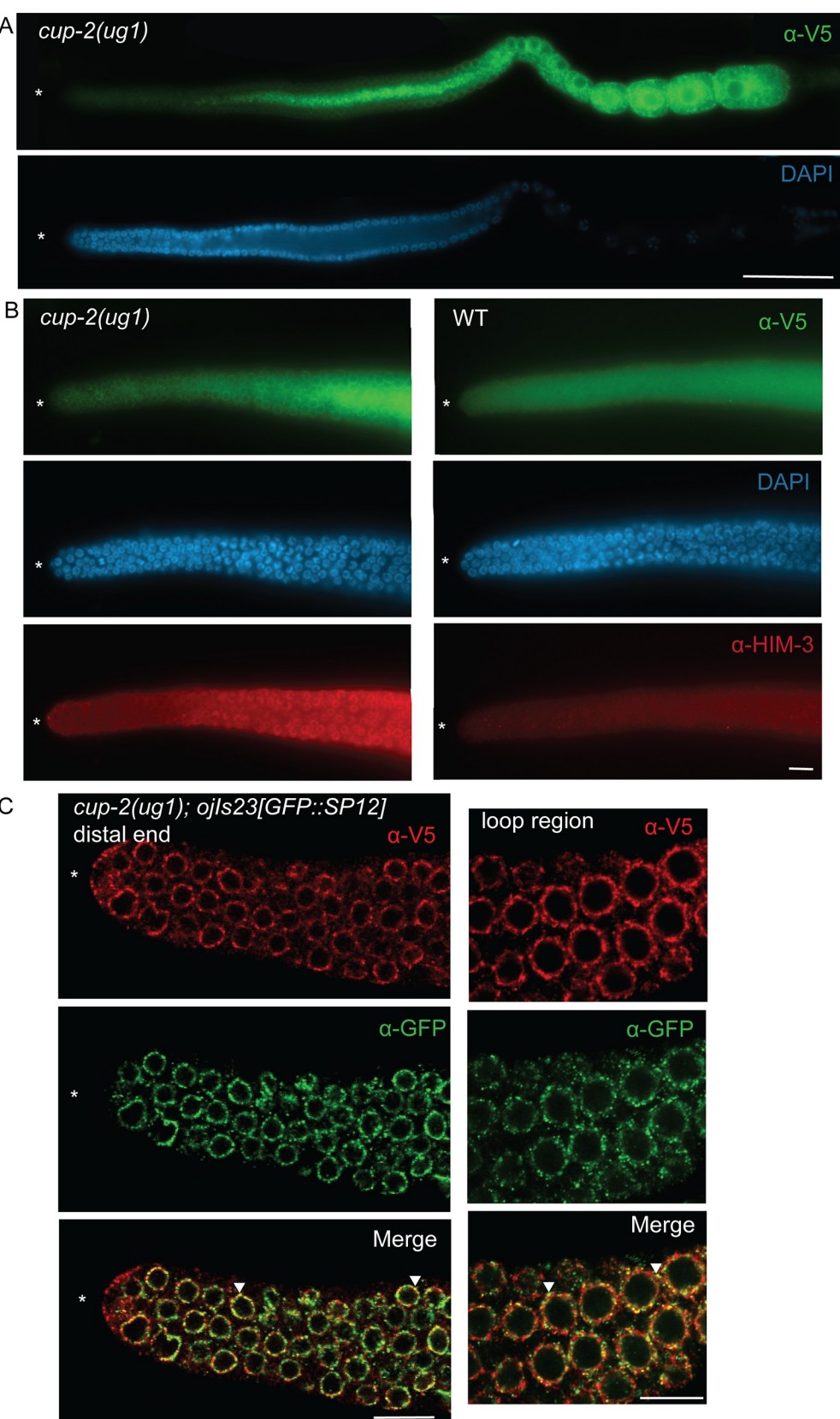

**Fig 5. CUP-2 is expressed in the proliferative zone of the germline and co-localizes with SP12, an ER marker.** A. CUP-2 expression visualized by α-V5 mouse immunostaining in the germline of *cup-2(ug1)*, a V5::2XFLAG tagged version of *cup-2*. DAPI staining was used to visualize nuclei. Asterisk, distal tip. Scale bar = 20μm. B. CUP-2 expression in the distal region of germline of *cup-2(ug1)* visualized by α-V5 mouse immunostaining in comparison to wild type untagged worms. Both genotypes were processed together and imaged on the same slide at the same intensity to enable comparison, except that *cup-2(ug1)* were preincubated α-HIM-3 antibodies to distinguish them from wild-type germlines. DAPI staining was used to visualize nuclei. Asterisk, distal tip. Scale bar = 10μm. C. Confocal images of CUP-2 expression in *cup-2(ug1); ojis23[GFP::SP12]* germline visualized by α-V5 rabbit immunostaining in distal and loop regions of the germline. Expression of ER marker SP12 was visualized by α-GFP mouse immunostaining. Images were acquired using a confocal microscope. Arrowheads point to regions of colocalization. Asterisk, distal tip. Scale bar = 10μm.

was less as compared to when using *cup-2* mutants, this is likely due to RNAi only partially reducing CUP-2 activity (Fig 6A and Table 4).

## *cup-2*'s role in retrograde transport may partially contribute to its function in promoting germ cell proliferation

Apart from functioning in ERAD, Derlins also function with SNX-1, a sorting nexin, in the retrograde transport of integral membrane proteins from the endosomes to the Golgi apparatus [43,52]. Our analysis of CUP-2's expression pattern revealed that some CUP-2 was present outside the ER, in the cytoplasm of germ cells (Fig 5C). It is possible that some of this cytoplasmic CUP-2 associates with the endosomal compartments for retrograde transport and may be responsible for *cup-2*'s role in promoting stem cell proliferation. Therefore, we asked whether CUP-2's role in retrograde transport could account for its function in promoting germ cell proliferation. We found that *snx-1(tm847)* suppresses the tumourous phenotype in some *puf-8(q725); glp-1(oz264gf)* animals. While 95% of the gonads of *puf-8(q725); glp-1 (oz264gf)* were completely tumourous, 74% of *puf-8(q725); glp-1(oz264); snx-1(tm847)* were completely tumourous, although none were wild-type (Fig 6B and Table 5). This level of suppression was much weaker than that observed when *cup-2* activity is absent, in which only 14% of the germlines are completely tumourous and 22% were wild-type (Fig 6B and Table 5). Therefore, we conclude that although retrograde transport may partially contribute to the loss of *cup-2* suppressing *glp-1(gf)* mediated overproliferation, it is unlikely to be the main mechanism by which *cup-2* acts in this role. There is an emerging link being established between the retromer complex and Notch signalling in other developmental contexts and the slight tumour suppression that we see in *snx-1* mutants could be due to the retromer complex directly affecting the trafficking of the Notch receptor [83–85].

## Chemical induction of ER stress suppresses GLP-1/Notch overproliferation, and the ability of *cup-2(0)* to suppress over-proliferation, requires *xbp-1*, a key player in the Unfolded Protein Response

We have demonstrated that loss of Derlin activity suppresses *glp-1(gf)* mediated overproliferation in the *C. elegans* germ line. Derlin proteins are known to have important functions in mediating ERAD [44,45,47,78,79,86,87]. In the absence of CUP-2 activity misfolded proteins accumulate, resulting in UPR induction [43,44]. Previous studies have demonstrated that *cup-2* mutants express high levels of *hsp-4::gfp*, a reporter for XBP-1-dependent UPR activation [43,44]. We hypothesized that the induction of ER stress and the UPR could be responsible for the tumour suppression seen in *cup-2* mutants. Therefore, we first asked whether inducing ER stress by another means (chemical induction) is sufficient to suppress Notch-dependent overproliferation in a manner similar to when Derlin function is reduced/eliminated.

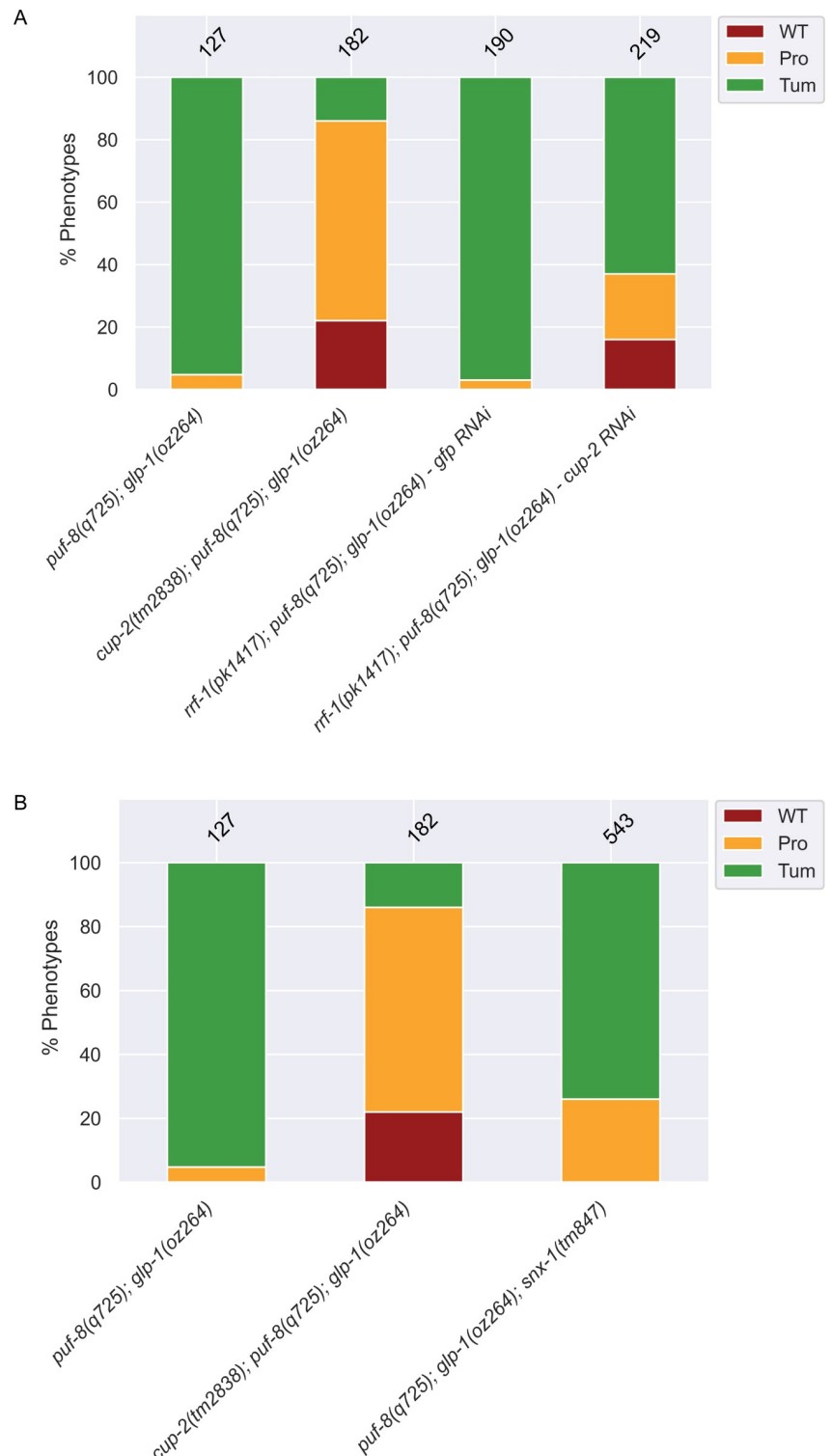

**Fig 6. Retrograde transport may contribute to *cup-2* activity and *cup-2* is required mainly within the germline to promote germ cell proliferation.** A. Quantification of phenotypic analysis of the effect of loss of *snx-1*, *cup-2*'s partner in retrograde transport, on suppression of *puf-8(q725); glp-1(oz264)* tumours. Phenotypes were analyzed by α-REC-8 and α-HIM-3 immunostaining. B. Quantification of phenotypic analysis of the effect of knockdown of *cup-2* activity within the germline by using a *rrf-1(pk1417)* mutant background which largely restricts RNAi efficacy to the germline. *cup-2* RNAi and *gfp* RNAi was performed by injection of dsRNA. *puf-8(q725); glp-1(oz264)* and *cup-2(tm2838); puf-8*

*(q725); glp-1(oz264)* phenotypic analysis is included for reference. Phenotypes were analyzed by α-REC-8 and α-HIM-3 immunostaining.

To induce ER stress, we used two commonly used chemicals, DTT and Thapsigargin (TG), at doses known to induce ER stress in worms [88,89]. TG is a specific inhibitor of an ER membrane $Ca^{2+}$-ATPase. By depleting the calcium stores of the ER, it alters the protein folding environment thereby inducing ER stress. DTT inhibits disulfide bridge formation in proteins, preventing proteins from folding properly and thereby inducing ER stress. We found that both DTT and TG suppress *puf-8(q725); glp-1(oz264gf)* overproliferation in a dose-dependent manner (Figs 7A and S5A and Tables 6 and S3). Of the two chemicals, DTT was the stronger suppressor as fewer complete tumours were seen at high DTT doses and some wild-type germlines were also observed, as compared to high TG doses where the reduction in complete tumours was smaller and no wild-type germlines were observed (Fig 7A and Table 6).

We have demonstrated that Derlin mutants partially suppress GLP-1/Notch-dependent tumours, such as *puf-8(q725); glp-1(oz264gf)*, but not GLP-1/Notch-independent tumours, such as *gld-2(0) gld-1(0)* tumours (Figs 2 and 3). We tested whether chemical induction of ER stress also specifically suppressed GLP-1/Notch-dependent tumours. We found that DTT treatment suppressed *glp-1(oz264gf)* and *glp-1(ar202gf)* in a dose-dependent manner at 25˚C (S5B Fig and S3 Table). While 28% of the gonads of 0 mM DTT treated *glp-1(oz264gf)* worms were completely tumourous, 17% of 2 mM DTT treated and only 11% of 5 mM DTT treated were completely tumourous (S5B Fig and S3 Table). Similarly, DTT treatment of *glp-1 (ar202gf)* worms at 25˚C resulted in dose-dependent suppression of tumours with 0 mM DTT treatment producing 26% completely tumourous gonads, while only 15% and 10% completely tumourous gonads were observed for the 2 mM and 5 mM DTT treatments, respectively (S5B Fig and S3 Table). We found that DTT treatment of worms carrying the weaker gain-of-function *glp-1(oz264gf)* allele resulted in more wild-type looking gonads (69%, 82% and 88% wild-type with 0 mM, 2 mM and 5 mM DTT treatments, respectively) rather than protumourous gonads (S5B Fig and S3 Table). The nature of the tumour suppression of the stronger gain-of-function *glp-1(ar202gf)* allele with DTT treatment was more variable and favoured the production of more wild-type gonads at the intermediate 2 mM dose (19% and 34% protumours for 0 and 2 mM DTT, respectively), but at the higher 5 mM dose more protumourous gonads were observed (55%, 51%, 79% Protumours with 0 mM, 2 mM and 5 mM DTT treatments,

**Table 4. Loss of *cup-2* activity in the germline is likely responsible for tumour suppression.**

| Genotype[19] | RNAi | WT[20] | Protumour[21] | Complete Tumour[22] | n[23] |
|---|---|---|---|---|---|
| *puf-8(0); glp-1(gf)* | none | 0% | 5% | 95% | 127 |
| *cup-2(0); puf-8(0); glp-1(gf)* | none | 22% | 64% | 14% | 182 |
| *rrf-1(0); puf-8(0); glp-1(gf)* | *gfp* | 0% | 3% | 97% | 190 |
| *rrf-1(0); puf-8(0); glp-1(gf)* | *cup-2* | 16% | 21% | 63% | 219 |

[19]Alleles used *puf-8(q725)*, *cup-2(tm2838)*, *rrf-1(pk1417)* and *glp-1(oz264gf)*

[20]Wild-type (WT) is defined as a gonad arm with distal α-REC-8(+) cells followed by α-HIM-3(+) cells and presence of both sperm and oocytes in the proximal arm of the gonad

[21]A protumour is defined as a gonad arm containing both α-REC-8(+) and α-HIM-3(+) cells, but not differentiated sperm and oocytes. Proximal tumourous gonad arms with only sperm but no oocytes were counted as protumours. Gonad arms with mostly only α-REC-8(+) and a few α-HIM-3(+) positive cells were also counted as protumourous.

[22]Complete tumour is defined as a gonad arm that contains only α-REC-8(+) cells and no α-HIM-3(+) cells

[23]Number of gonad arms

**Table 5. Loss of *snx-1* weakly suppresses tumours.**

| Genotype[24] | WT[25] | Protumour[26] | Complete Tumour[27] | n[28] |
|---|---|---|---|---|
| *puf-8(0); glp-1(gf)* | 0% | 5% | 95% | 127 |
| *puf-8(0); glp-1(gf); snx-1(0)* | 0% | 26% | 74% | 543 |
| *cup-2(0); puf-8(0); glp-1(gf)* | 22% | 64% | 14% | 182 |

[24]Alleles used *puf-8(q725), cup-2(tm2838), snx-1(tm847))* and *glp-1(oz264gf)*

[25]Wild-type (WT) is defined as a gonad arm with distal α-REC-8(+) cells followed by α-HIM-3(+) cells and presence of both sperm and oocytes in the proximal arm of the gonad

[26]A protumour is defined as a gonad arm containing both α-REC-8(+) and α-HIM-3(+) cells, but not differentiated sperm and oocytes. Proximal tumourous gonad arms with only sperm but no oocytes were counted as protumours. Gonad arms with mostly only α-REC-8(+) and a few α-HIM-3(+) positive cells were also counted as protumourous.

[27]Complete tumour is defined as a gonad arm that contains only α-REC-8(+) cells and no α-HIM-3(+) cells

[28]Number of gonad arms

respectively) background, suggesting that the tumour suppression was weaker in the stronger gain-of-function *glp-1(ar202gf)* DTT treated worms as compared to weaker gain-of-function *glp-1(oz264gf)* DTT treated worms (S5B Fig and S3 Table). However, DTT treatment was unable to suppress GLP-1/Notch-independent *gld-2(0) gld-1(0)* tumours at all (S5C Fig and S4 Table). Therefore, chemical induction of ER stress mimicked the effect of loss of Derlin activity in suppressing overproliferation, suggesting that ER stress and the consequent induction of the UPR may be the mechanism by which Derlin mutants are able to suppress Notch-dependent tumours.

Previous studies demonstrated that *cup-2* mutants are synthetically lethal with *ire-1*, the sensor of the IRE-1/XBP-1 branch of the UPR [43,49], supporting the suggested role of *cup-2* in ERAD. Additionally, other ERAD; *ire-1* and ERAD; *xbp-1* double mutants fail to induce the UPR (Sasagawa et al. 2007), suggesting dependence on the IRE-1/XBP-1 pathway. To further our understanding of how induction of the UPR suppresses GLP-1/Notch mediated overproliferation we asked if XBP-1 is necessary for the *cup-2* mediated suppression of overproliferation. We found that RNAi against *xbp-1* in *cup-2(tm2838); puf-8(q725); glp-1(oz264)* increased the proportion of tumourous worms (Fig 7B and Table 7). Therefore, this suggests that the suppression of GLP-1/Notch mediated overproliferation by reducing *cup-2* activity requires *xbp-1* activity and UPR induction.

We have demonstrated that *cup-2(0); der-2(0)* double mutants suppress GLP-1/Notch tumours more strongly than either single mutant. In addition, a previous study has demonstrated that *cup-2(0); der-2(0)* mutants more strongly induce the UPR than either single mutant [43]. Therefore, we wondered whether further activating the UPR by increasing ER stress in a *cup-2(tm2838); puf-8(q725); glp-1(oz264)* background could increase the tumour suppression even further. We found that inducing ER stress by DTT treatment in the *cup-2(tm2838); puf-8(q725); glp-1(oz264)* background increased the tumour suppression to such an extent such that no tumourous worms were observed at the highest DTT dose of 5 mM tested (Fig 7C and Table 6). This suggests that in *cup-2* mutants, although the UPR is activated and is able to suppress GLP-1/Notch dependent tumours, increasing the level of UPR activation even further results in stronger suppression of overproliferation. Therefore, the UPR is capable of being induced more than that achieved through loss of *cup-2* activity, and increasing UPR induction increases the suppression of *glp-1(gf)* induced overproliferation (Fig 7C).

We wanted to test whether chemical induction of ER stress alone, without disturbing *cup-2* activity, was sufficient to suppress GLP-1/Notch signalling levels. We therefore measured the

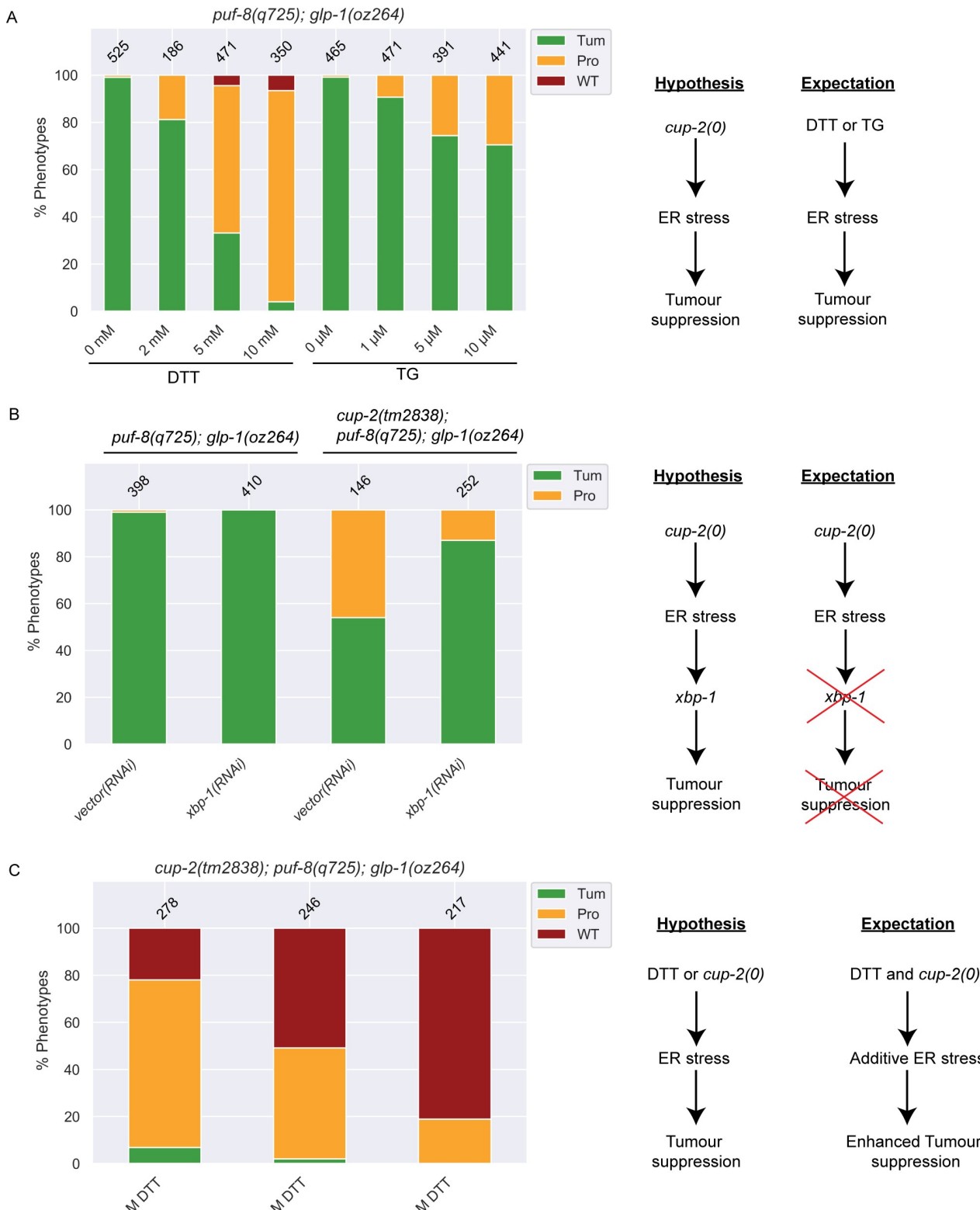

**Fig 7. Chemical induction of ER stress also suppresses Notch-dependent tumours and suppression of tumours by loss of *cup-2* activity requires *xbp-1*, a key player in the Unfolded Protein Response.** A. Quantification of phenotypic analysis of the effect of increasing doses of DTT and Thapsigargin (TG) on suppression of *puf-8(q725); glp-1(oz264)* tumours. We hypothesize that if ER stress induced by abrogating *cup-2* activity is responsible for the suppression of the tumour, then inducing ER stress by chemical means should also produce a similar effect. Phenotypes were

analyzed by whole mount DAPI staining. B. Quantification of phenotypic analysis of the effect of *xbp-1* knockdown by RNAi on *puf-8(q725); glp-1 (oz264)* tumours. We hypothesize that if activation of the branch of the UPR is involved in the response to ER stress and suppression of tumours in *cup-2(tm2838)* worms, then knocking down *xbp-1* function in *cup-2(tm2838); puf-8(q725); glp-1(oz264)* would be expected to reverse the tumour suppression seen back to more tumourous gonads. RNAi was performed by feeding. RNAi on *puf-8(q725); glp-1(oz264)* was performed as a control. Phenotypes were analyzed by whole mount DAPI staining. C. Quantification of phenotypic analysis of the effect of inducing ER stress by DTT treatment on *cup-2(tm2838); puf-8(q725); glp-1(oz264)* tumours. We hypothesize that if either DTT or loss of *cup-2* function can induce ER stress and suppress the tumours, then the combination of the two would enhance the suppression. Phenotypes were analyzed by whole mount DAPI staining.

protein levels of the readout for GLP-1/Notch signalling, SYGL-1, in worms treated with DTT. We found that in both wild-type and *glp-1(ar202)* worms, increasing doses of DTT lead to a more distal shift of the peak of SYGL-1 expression curve in the germline, accompanied by shortening of the average position of the transition zone (Figs 8 and S6). This distal shift of the SYGL-1 expression curve is reminiscent of the effect of removing *cup-2* activity in *glp-1(ar202)* gonads (Fig 4B). Taken together, this supports the possibility that induction of ER stress by removal of *cup-2* activity and the consequent induction of the UPR may be responsible for suppressing GLP-1/Notch signalling levels and Notch-dependent overproliferation. We further found that knockdown of *xbp-1* activity reverses the distal shift of the SYGL-1 expression curve in *cup-2(0); glp-1(ar202)* gonads, suggesting that a functional UPR is indeed needed to suppress excessive GLP-1/Notch signalling levels when *cup-2* activity is removed (S7 Fig).

## Discussion

We have shown that Derlin mutants suppress germline tumours caused by gain-of-function mutations affecting the Negative Regulatory Region (NRR) in GLP-1/Notch's extracellular domain [32,90,91] and in proportion to the relative strength of the gain-of-function mutation. The suppression of these tumourous phenotypes is strongest in animals lacking both *cup-2* and *der-2* function. Derlin mutants do not suppress Notch-independent tumours. Even in mutants that have some cells that enter into meiosis, such as the *gld-2(0) gld-1(0)* Notch-independent tumour, loss of both *cup-2* and *der-2* function fails to suppress the tumour. This

**Table 6. DTT and Thapsigargin (TG) suppress *glp-1(gf)* overproliferation in a dose-dependent manner.**

| Treatment | Genotype[29] | WT[30] | Protumour[31] | Complete Tumour[32] | n[33] |
|---|---|---|---|---|---|
| 0 mM DTT | *puf-8(0); glp-1(gf)* | 0% | 1% | 99% | 525 |
| 2 mM DTT | *puf-8(0); glp-1(gf)* | 0% | 19% | 81% | 186 |
| 5 mM DTT | *puf-8(0); glp-1(gf)* | 4% | 62% | 33% | 471 |
| 10 mM DTT | *puf-8(0); glp-1(gf)* | 7% | 89% | 4% | 350 |
| 0 μM TG | *puf-8(0); glp-1(gf)* | 0% | 1% | 99% | 465 |
| 1 μM TG | *puf-8(0); glp-1(gf)* | 0% | 9% | 91% | 471 |
| 5μM TG | *puf-8(0); glp-1(gf)* | 0% | 26% | 74% | 391 |
| 10 μM TG | *puf-8(0); glp-1(gf)* | 0% | 29% | 71% | 441 |
| 0mM DTT | *cup-2(0); puf-8(0); glp-1(gf)* | 22% | 71% | 7% | 278 |
| 2mM DTT | *cup-2(0); puf-8(0); glp-1(gf)* | 51% | 47% | 2% | 246 |
| 5mM DTT | *cup-2(0); puf-8(0); glp-1(gf)* | 81% | 19% | 0% | 217 |

[29]Complete genotypes *puf-8(q725); glp-1(oz264)* and *cup-2(tm2838); puf-8(q725); glp-1(oz264)*

[30]Wild-type (WT) is defined as a gonad arm with presence of both sperm and oocytes in the proximal arm of the gonad as seen by whole mount DAPI staining

[31]A protumour is defined as a gonad arm with a mass of proliferative cells in the proximal end preceded more distally by presence of sperm and/or eggs as seen by whole mount DAPI staining

[32]Complete tumour is defined as a gonad arm that contains only proliferative cells as seen by whole mount DAPI staining

[33]Number of gonad arms

**Table 7. Reduction in *xbp-1* function partially enhances *glp-1(gf)* overproliferation when also lacking *cup-2* activity.**

| Genotype | Treatment | WT[34] | Protumour[35] | Complete Tumour[36] | n[37] |
|---|---|---|---|---|---|
| *puf-8(0); glp-1(gf)*[38] | *vector(RNAi)* | 0% | 1% | 99% | 398 |
| | *xbp-1(RNAi)* | 0% | 0% | 100% | 410 |
| *cup-2(0); puf-8(0); glp-1(gf)*[39] | *vector(RNAi)* | 0% | 46% | 54% | 146 |
| | *xbp-1(RNAi)* | 0% | 13% | 87% | 252 |

[34]Wild-type (WT) is defined as a gonad arm with presence of both sperm and oocytes in the proximal arm of the gonad as seen by whole mount DAPI staining

[35]A protumour is defined as a gonad arm with a mass of proliferative cells in the proximal end preceded more distally by presence of sperm and/or eggs as seen by whole mount DAPI staining

[36]Complete tumour is defined as a gonad arm that contains only proliferative cells as seen by whole mount DAPI staining

[37]Number of gonad arms

[38]Complete genotype *puf-8(q725); glp-1(oz264)*

[39]Complete genotype *cup-2(tm2838); puf-8(q725); glp-1(oz264)*

suggests that the tumour suppression is specific to Notch signaling. By measuring the expression levels and extent of SYGL-1, a readout for GLP-1/Notch signalling, we found that the extent/region of SYGL-1 expression is compressed in *cup-2; glp-1(ar202gf)* germlines compared to that of *glp-1(ar202gf)* germlines, suggesting that the suppression of overproliferation in *glp-1(gf)* mutants by the reduction of Derlin function is likely achieved by reducing GLP-1/Notch signaling levels.

We found that many of the key features of germline tumour suppression seen in Derlin mutants are phenocopied by chemically inducing ER stress. Consistent with the suppression observed with the reduction of Derlin activity, inducing ER stress suppresses Notch-dependent tumours but not Notch-independent tumours, and reduces the extent of the SYGL-1 expression in *glp-1(ar202gf)* germlines. We further found that *cup-2*'s suppression of a *puf-8(0); glp-1 (oz264gf)* Notch-dependent tumour requires *xbp-1* activity, which encodes a component of the IRE-1/XBP-1 branch of the Unfolded Protein Response. Moreover, we found that increasing ER stress even further by chemical treatment in *cup-2(0); puf-8(0); glp-1(oz264gf)* worms results in increased tumour suppression. Our results support a model in which ER stress, in general, counteracts the development of Notch-dependent germline tumours and suggest that the IRE-1/XBP-1 branch of UPR may be one of the arms involved in this counterbalance.

## Reduction of Derlin activity reduces Notch signaling in gain-of-function mutants

We have demonstrated that a reduction in Derlin function reduces the amount of overproliferation observed in various *glp-1* gain-of-function mutants, but fails to suppress overproliferation due to a loss of *gld-1* and *gld-2* pathway genes, even in mutants that have a number of cells that enter meiotic prophase. Derlins function, in part, by shuttling misfolded proteins from the ER to the cytosol for degradation, as part of ERAD [44,45,47,78,87,92–95]. Therefore, the suppression of germline overproliferation in *cup-2* and *der-2* mutants is likely the result of an accumulation of unfolded or misfolded proteins. There are likely many models as to how this accumulation could reduce the level of GLP-1/Notch signaling and the level of overproliferation. For example, this accumulation could cause a core member of the Notch signaling pathway, or a positive regulator of Notch signaling, to not function properly. However, if a reduction in Derlin function were to interfere with a core member or a positive regulator of GLP-1/Notch signaling, then we would expect to have seen a reduction in GLP-1/Notch signaling, as determined by SYGL-1 accumulation, when Derlin function was reduced in an

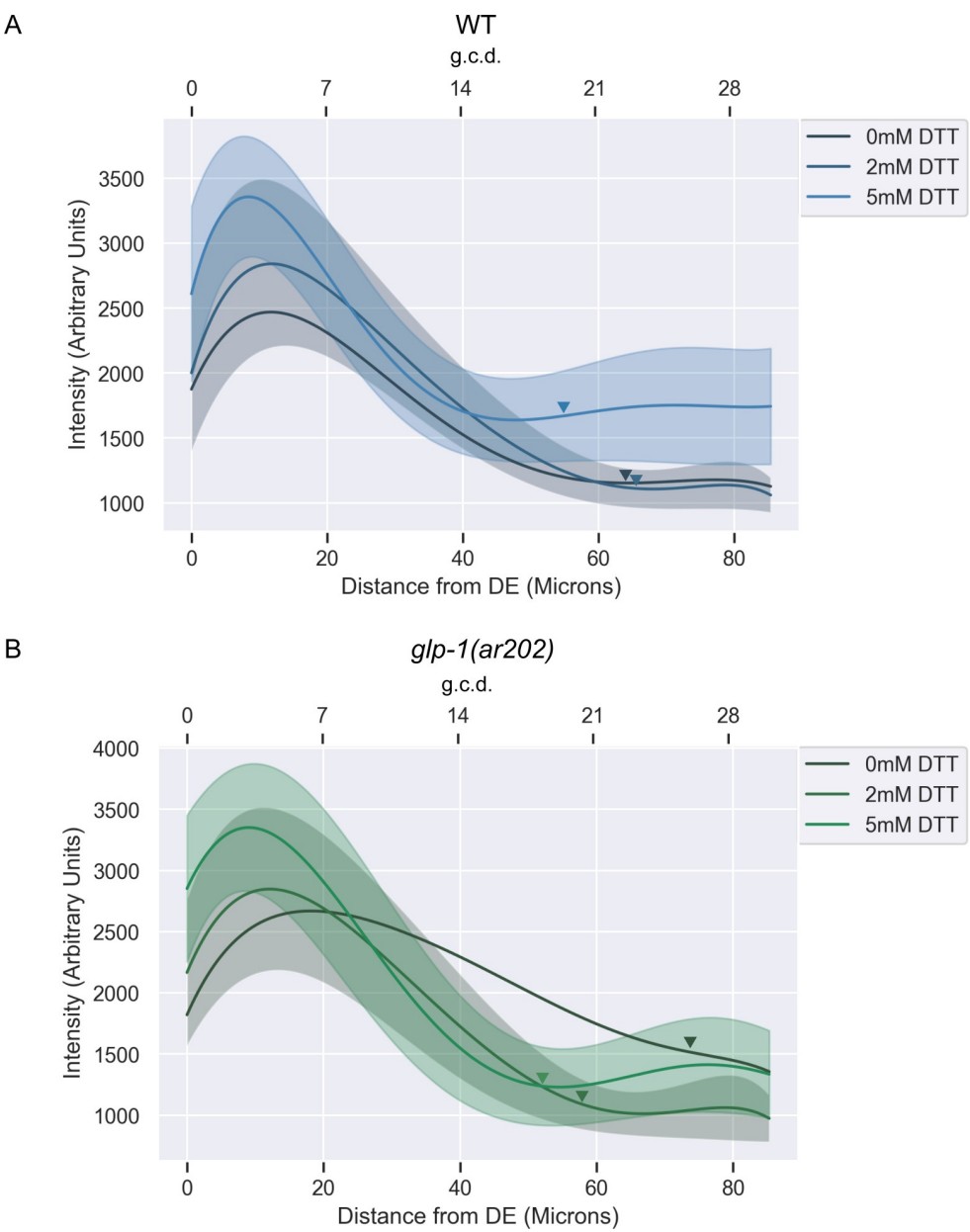

**Fig 8. Chemical induction of ER stress by DTT reduces the spread of SYGL-1, a readout for GLP-1/Notch signalling in *glp-1(ar202)* germlines.** A. Normalized, fitted average SYGL-1 intensities of otherwise wild type gonads harbouring the *sygl-1(am307)* allele grown on DTT. Shaded areas indicate unscaled fitted standard deviation of the intensity measurements for each genotype. Standard deviation for average 0 mM DTT treatment has not been shown for ease of visualization but can be seen in S6 Fig. Average normalized intensities and standard deviations were fit to a sixth order polynomial. Fifteen gonads were analyzed for intensity measurements. Arrowheads point to the average location of the transition zone measured in at least seven gonads of each genotype. Distances from distal end (DE) were measured in microns and converted to germ cell diameters (g.c.d) as a reference, by assuming 1 g.c.d. = 2.833 microns. B. As in (A) but including the *glp-1(ar202)* allele.

otherwise wild-type background. However, we only see dramatic reduction of Notch signaling in *glp-1* gain-of-function mutants.

It is intriguing that proper protein folding of the Notch receptor has been shown to be important for the function of the NRR (Negative Regulator Region), where many of the

known Notch gain-of-function mutations cluster [31,32,96–98]. Folding in this region is thought to protect the S2 cleavage site, allowing it only to be accessed upon ligand binding [99–101]. Improper folding caused by mutations in this region are thought to allow for ligand-independent cleavage and activation [100,102]. Could reduction in ERAD affect the folding of the GLP-1/Notch receptor, including the NRR region, and thereby affect the level of Notch signaling? We have demonstrated that a reduction in Derlin activity reduces the increased level of Notch signaling achieved by *glp-1* gain-of-function mutants. Obviously, GLP-1/Notch proteins harbouring gain-of-function mutations must not be misfolded to a degree in which all such proteins are detected by the ERAD system and degraded, otherwise they would not be available for ligand independent signaling. However, perhaps some proportion of the gain-of-function proteins are misfolded to the point of being detected by the ERAD system and targeted for degradation. When Derlin activity is reduced, these and other misfolded proteins may accumulate in the ER, causing ER stress and the activation of UPR. UPR activation causes the initiation of signaling pathways meant to reduce ER stress, including enhancing protein folding and degradation [103]. Therefore, it is possible that activation of UPR could result in enhanced protein folding of key factors in the Notch signaling pathway, or increased degradation of misfolded proteins, perhaps including the aberrant gain-of-function form of the GLP-1/Notch receptor itself. Indeed, enhanced protein folding could result in the NRR region properly folding, even if harbouring a gain-of-function mutation, thereby mimicking a more wild-type form of the protein in terms of activity. Indeed, the *glp-1(ar202)* gain-of-function mutation that we demonstrated was suppressed most strongly by a reduction in Derlin activity does still have some ligand-dependence [104], suggesting that either the NRR domain in these proteins is not as misfolded as in other gain-of-function mutants, or that the proportion of misfolded proteins is lower. In either situation, enhancement of proper NRR folding would suppress the increase in GLP-1/Notch signaling levels. Alternatively, it is possible that UPR induction enhances the recognition of GLP-1 gain-of-function proteins as aberrantly folded, leading to enhanced degradation of the receptor. Enhanced degradation would reduce the overall levels of GLP-1 receptors, thereby resulting in a decrease in signalling levels. It is also possible that the models of enhanced degradation and/or enhanced folding could apply to a regulator of GLP-1/Notch signalling that interacts with the NRR of the GLP-1 receptor; however, we consider this less likely given our understanding of how other components of the ERAD machinery interact with Notch signalling, as explained below.

## ERAD, UPR and Notch signaling

The Derlins are not the first components of the ERAD machinery to be identified as interacting with Notch signaling in *C. elegans*, perhaps hinting at some inherent sensitivity of this signalling pathway to ERAD-driven measures for protein quality control. SEL-11 and SEL-1, homologs of the mammalian HRD1 and SEL1 ERAD components [105,106], were initially identified in a screen for suppressors of the egg-laying defective phenotype (Egl) of a partial loss-of function allele of *lin-12* [107], which, like *glp-1*, encodes a homologue of the Notch protein [108–110]. SEL-11/HRD1/Hrd1p is an E3 ubiquitin ligase that ubiquitinates misfolded or unfolded proteins, while SEL-1/SEL1L/Hrd3p interacts with SEL-11/HRD1/Hrd1 [13,78,94,111–114]. In *C. elegans*, *sel-1* and *sel-11* mutations were also found to suppress the maternal effect embryonic lethality phenotype of *glp-1 (e2142)*, a partial loss-of-function allele, suggesting that *sel-1* and *sel-11* interact with Notch signaling rather than with processes regulated by Notch [107,115]. Importantly, loss of *sel-1* and *sel-11* strongly suppresses *lin-12* partial-loss-of-function alleles, but do not suppress a *lin-12* null, suggesting that the LIN-12/Notch receptor must be present, even in a mutant form, in order for suppression to occur

[106,107,115]. While *sel-1* and *sel-11* suppress *glp-1* and *lin-12* partial loss-of-function alleles, and therefore are thought to be negative regulators of Notch signaling, we have demonstrated that reduction in Derlin activity, through *cup-2* and/or *der-2* mutations, results in suppression of *glp-1* gain-of-function alleles, and has little if any effect on *glp-1* partial loss-of-function alleles, suggesting that the Derlins are positive regulators of Notch signaling.

Derlins are thought to complex with SEL-11/HRD1 and SEL-1/SEL1L to facilitate transport of misfolded proteins from the ER to the cytosol, where they would be degraded by the proteasome [78,116–118]. A recent study in yeast, suggests that Der1 and Hrd1 form two "half-channels" to allow ERAD substrates to move through the ER membrane [118]. Therefore, it would be anticipated that a reduction of Derlin function would have the same overall consequence as a reduction in SEL-1/SEL1 and SEL-11/HRD1 function in reducing the ability of unfolded or misfolded proteins to be degraded. However, we have demonstrated that reduction of Derlin activity has an opposite effect on Notch signalling as that described for SEL-1/SEL1 and SEL-11/HRD1 [106,107]. This may reveal that, at least in some contexts, the SEL-11/HRD1 complex and Derlins may have opposite effects on ERAD, or perhaps that removing individual components of the complex leads to different compensatory mechanisms with opposing outcomes on Notch as a substrate.

However, we believe this apparent contradiction may be resolved by considering the folding properties of the mutant Notch alleles in which *sel-1* and *sel-11* have been characterized. Perhaps the *glp-1* and *lin-12* partial loss-of-function mutants used in these studies are also recognized by the ERAD machinery as misfolded. If so, then upon the loss of ERAD factors, the UPR would be induced and these mistakenly folded receptors could be restored to wild-type receptor activity levels through the enhanced degradation and/or enhanced folding mechanisms proposed above. Such a model also accommodates the need for some mutant form of the Notch receptor to be present for the *sel-1* and *sel-11* mutants to be able to exert their influence on Notch signalling [107,115]. Therefore, we speculate that loss of ERAD factors leads to correction of aberrantly folded mutant Notch receptors, since the UPR does not differentiate between whether a receptor is gain-of-function or loss-of-function in activity, but instead simply recognizes them as being misfolded and needing correction. Some support for such a model in which the UPR serves as a quality control mechanism for aberrant Notch folding and signalling, exists amongst studies using Drosophila. While ectopic induction of ER stress alone has no effect on wildtype Notch receptor localization or levels, upregulation of the UPR or overexpression of a Notch-specific chaperone in a Notch-deficient mutant can restore Notch signalling levels in Drosophila [119,120].

## The extracellular portion of the GLP-1/Notch receptor may be more sensitive to ERAD defects

We recognize that the proposed model of the UPR as a quality control mechanism to correct aberrant Notch receptors would not apply to all mutant forms of Notch receptors, as we were unable to see strong enhancement of the *glp-1(bn18)* loss-of-function Glp phenotype in *cup-2* mutants, and the *sel-1* and *sel-11* mutants similarly fail to suppress the two reduction-of-function alleles *glp-1(q231)* and *glp-1(e2144)* [107]. However, it is intriguing that *glp-1(bn18)*, *glp-1(q231)* and *glp-1(e2144)* mutations all affect the intracellular portion of the GLP-1 receptor [63], while the *glp-1(ar202)*, *glp-1(ar224)*, *glp-1(oz264)* mutations, with which *cup-2* interacts, all affect the extracellular portion of the GLP-1 receptor [32,90,91]. Similarly, *glp-1(e2142)*, the partial loss-of-function allele with which *sel-1* and *sel-11* interact, also affects the extracellular portion [63,107]. Therefore, we speculate that the extracellular portion of the GLP-1/Notch receptor may be more sensitive to defects in components of the ERAD machinery than the

intracellular portion. This may suggest that ERAD/UPR specifically acts at the level of GLP-1/Notch receptor activation. Therefore, it is possible that ERAD/UPR could regulate the levels or activities of membrane-localized enzymes that process the GLP-1/Notch receptor.

### Derlins as a therapeutic target

Dysregulation of Notch signaling has been implicated in the development of many diseases, including certain cancers [121]. While some cancers are associated with loss-of-function mutations in the Notch receptor, others, such as Triple-Negative Breast Cancer, correlate with gain-of-function mutations, including those due to mutations in the NRR [121,122]. In fact, more than 50% of cases of human T cell acute lymphoblastic leukemia (T-ALL) are associated with activating mutations in NOTCH1, with the majority of these affecting the NRR [97]. Therefore, inhibition of Notch signaling is being actively pursued as a possible treatment for certain cancers [123]. However, since the Notch signaling pathway is utilized in the development and maintenance of so many tissues throughout the body, Notch inhibitors can have detrimental effects, such as intestinal toxicity [123]. Here we demonstrate that reduction in Derlin activity suppresses GLP-1/Notch gain-of-function mutants, but has little effect on normal Notch signaling as measured by SYGL-1 expression in *cup-2* mutants. Therefore, targeting Derlin activity in individuals suffering from diseases caused by Notch gain-of-function mutations, perhaps particularly those mutations that affect the extracellular portion of Notch, may provide a means to specifically target the aberrant Notch gain-of-function allele without causing adverse side effects associated with reducing systemic Notch signalling levels.

## Materials and methods

### Strains and genetics

All the strains used in this study were maintained at 20˚C unless otherwise indicated (S5 Table). The wildtype strain used was the N2 Bristol strain.

### Worm synchronization

Gravid adults were washed off of plates with PBS and washed three times with PBS to remove excess bacteria. Samples were then incubated with freshly prepared bleach solution (20% sodium hypochlorite, 50mM NaOH) for 4–5 minutes with vigorous vortexing. Excess bleach was washed off by washing the egg pellet with PBS three times. Eggs were allowed to hatch in PBS at 20˚C for 1–2 nights to obtain a synchronized population of L1 larvae.

### Dissections and immunostaining

Unless indicated otherwise in the figure legend, all phenotypes were analyzed in dissected gonads. About 150 synchronized adults were picked for dissections and dissected as previously described [65]. Dissected gonads were fixed in 3% paraformaldehyde for 10 minutes at room temperature, post-fixed using 100% methanol and kept at -20˚C for at least one night. Samples were then rehydrated by washing with PBT (PBS + 0.1% Tween-20) three times and blocked in 3% BSA at 4˚C for at least one hour. Primary antibodies used were α-REC-8 rat (1:200)[60], α-HIM-3 rabbit (1:750)[61], α-FLAG mouse M2 (1:1000) Sigma #F1804, α-V5 mouse (1:2000) Invitrogen #R960-25, α-V5 rabbit (1:1000) Cell Signaling Technology #D3H8Q and α-GFP mouse (3E6) (1:750) Molecular Probes #A11120. Primary antibodies were diluted in 3% BSA and incubated for at least 1 hour at room temperature. Secondary antibodies used were: α-Rat Alexa 488 (1:200) Molecular Probes #A21208, α-Rabbit Alexa 594 (1:500) Molecular Probes #A21207 and α-mouse Alexa 488 (1:200) Molecular Probes #A21202. Secondary antibodies

were diluted in 3% BSA and incubated for at least 2 hours at room temperature. Samples were washed three times with PBT with 5 min incubations, after primary and secondary antibody incubations to remove excess unbound antibodies. For visualizing nuclei, samples were incubated with 100ng/mL DAPI (4', 6- diamidino-2- phenylindole dihydrochloride) diluted 1:1000 in PBS for 5 minutes. Stained gonads were visualized by mounting on a 1% agarose pad.

### Image acquisition

All images were acquired on a Zeiss Imager Z.1 microscope fitted with an AxioCam MRm camera using AxioVision 4.8.2.0 software. Z stacks were taken at 1μm intervals. Confocal images were taken on a Leica SP5 laser confocal microscope. For experiments measuring SYGL-1 levels, distal ends of gonads were focused in the middle focal plane, as much as possible, for image acquisition.

For determining CUP-2 localization by comparing α-V5 staining intensity in XB681 *cup-2 (ug1)* vs N2 dissected gonads, the two strains were fixed, permeabilized and blocked independently and in parallel. The XB681 sample was incubated with α-HIM-3 antibody while the N2 sample was kept in block overnight. This helped mark the XB681 gonads so that they can be easily distinguished when mixed with the N2 sample. The next day, the XB681 sample was washed three times with PBT and mixed with the N2 sample so that subsequent processing on the two samples could be done in the same tube to enable a direct comparison. The mixed sample was incubated with α-V5 mouse overnight. The next day the sample was processed with secondary antibodies as described above and mounted on the same pad. Exposure measurements were taken in three positions across the slide and the average exposure was set as the exposure for acquisition of images across the whole slide. Therefore, both XB681 and N2 samples were imaged with the same exposure settings.

### Image processing and analysis

Fiji (Fiji is Just ImageJ) was used to process and analyze images [124,125]. Brightness and contrast adjustments, and the addition of scale bars was done in Fiji. Images were cropped and organized into figure panels using Adobe Illustrator. For stitching a complete gonad picture, the stitching plugin in Fiji was used [126]. For counting the total number of cells in the proliferative zone, Z stack images of the distal ends of gonads were manually analyzed using the Cell Counter plugin [127].

Statistical analysis and plotting was done in Python using Seaborn, a visualization library based on the 2D graphics package, Matplotlib [128,129]. t-tests (independent) were performed using the statannot package [130]. Chi-square tests were performed using SciPy [131].

### SYGL-1 intensity measurements

For an experiment, the strain of interest was dissected in parallel with WU1770 *sygl-1(am307)* as an internal positive control. The two strains were fixed, permeabilized and blocked independently. The WU1770 dissected samples were incubated with α-HIM-3 antibody while the other sample was left in block overnight. The next day, the WU1770 sample was washed three times with PBT and the two samples were pooled for subsequent processing with α-FLAG antibody, secondary antibody, DAPI and mounting.

For each experiment, the ideal exposure settings for each channel was determined by sampling three germlines of each genotype across the slide using autoexposure and then using the average exposure measurement as the setting for acquiring images for that slide. Germlines were focussed in the central plane, through the middle of the rachis for acquiring images. For analyses, pictures of 15 gonads of each genotype were analyzed. The images were opened on

FIJI and a segmented line was drawn in the middle of the germline starting from the distal-most germ cell to the most proximal cell in the field of view to generate a plot profile for each germline. The values from the plot profile were copied into excel, subsequent analyses i.e., calculation of the average intensity of each genotype, standard deviation, normalization and plotting were done using Seaborn. For comparing multiple genotypes against each other, normalization was performed to the average WU1770 plot profiles for these experiments. For example, when comparing WU1770, XB709, XB710 and XB711 genotypes against each other, the 'scale average' was calculated as the average WU1770 plot profile of these three experiments (namely XB709 vs WU1770, XB710 vs WU1770 and XB711 vs WU1770). To obtain the scaling factor for XB709, the 'scale average' was divided by the average WU1770 plot profile for the XB709 experiment. The scaling factor for XB709 was then multiplied by the plot profile for XB709 to normalize it. The same was repeated for the XB710 and XB711 experiments to normalize their plot profiles. The normalized plot profiles of XB709, XB710 and XB711 were plotted on the same graph along with the 'scale average' to represent the average WU1770 intensity across these three experiments. In order to more easily contrast the plot profiles against each other, the plot profiles were fitted to a sixth-order polynomial to generate smoothened plots (Fig 4B). The raw, non-normalized plots for the three experiments is shown in S4 Fig. Since standard deviations could not be normalized, the standard deviation shown in Fig 4B is the same as that shown in S4 Fig. For determining the distance to the transition zone, a new line was drawn from the distal end to the first transition zone cells which were discernable by their crescent-shaped DAPI stained nuclei. Since the pictures were acquired in the central Z plane which has fewer nuclei, transition zones for all gonad pictures was difficult to determine. However, transition zones for at least seven images for each genotype was determined. The average distance to transition zone for each genotype was calculated in Excel and an arrowhead was drawn in Adobe Illustrator to indicate the average position on the plot. By measuring the distance to transition zone and using the cell counter plug in in FIJI for counting the number of germ cell diameters to transition zone for one WU1770 image, it was determined that one germ cell diameter (g.c.d.) approximately corresponded to 2.833 microns. This formula was applied to the distance in microns axes in the plots in Seaborn to convert the distance in microns to g.c.d. and a second X axis indicating the approximate distance in g.c.d. was generated as a reference.

## CRISPR/Cas9 editing

Two guide RNA sequences towards the 3'end of the *cup-2* coding region were selected manually by scanning for nearby NGG sites and running the seed sequence through NCBI BLAST to check that it was not predicted to match any other region of the genome. MfeI and XbaI restriction enzyme sites flanked 320bp fragments containing the sgRNA sequence were synthesized by Eurofins MWG Operon as per a previous study [72]. The plasmid was cut by MfeI and XbaI restriction enzymes and the 320bp fragment was cloned into a PU6::sgRNA vector derived from Addgene plasmid #46169, as per a previous study [72]. This generated PU6::*cup-2*_sgRNA_#4 (pDH386) and PU6::*cup-2*_sgRNA_#5 (pDH387) vectors for injections. These sgRNA plasmids were each used at the recommended 50ng/µl final concentration for injections [72].

For the repair template, a 258bp DNA fragment was designed containing a 90bp sequence corresponding to a codon optimized V5::2XFLAG tag placed upstream of the endogenous *cup-2* stop codon and flanked by homology sequences, based on recommendations for PCR directed repair templates by a previous study [73]. A silent point mutation (CAA to CAg) was introduced one base pair upstream of the PAM sequence recognized by sgRNA#5 to prevent potential re-cutting as advised by previous studies [73–75]. This repair template was

synthesized by Eurofins MWG Operon to generate plasmid pDH373. For injections, PCR using E10 (5'-ATCAGAGGAGCACGACAGCA-3') and E11 (5'-AGGAAAAAGGAAATAAATTA-3') primers on pDH373 as a template generated a smaller fragment that was cleaned up by QIAGEN MinElute PCR Purification Kit (#28004). The PCR fragment was used at 72ng/µl final concentration for injections. All other components of the injection mixture were used as previously described for co-conversions [74]. All plasmids used for injections were purified using QIAGEN Plasmid Midi Kit #12143. The *dpy-10* targeting sgRNA, pJA58 (Addgene #59933), was used at a final concentration of 25ng/µl. The *dpy-10(cn64)* oligo repair template was synthesized by IDT as 4nm Ultramer DNA Oligo and was used at 0.5µM final concentration for injections. The Cas9 expressing plasmid, pDD162 (Addgene #47549), was used at a final concentration of 50ng/µl [76]. Injected worms were placed on individual plates after injections and Dpy and/or Roller progeny were cloned individually and screened by PCR for insertions.

### dsRNA injections for RNAi

1000ng of a miniprep of full-length *cup-2* cDNA cloned into pL4440 and GFP cloned into pL4440 was used to PCR amplify dsDNA using a T7 primer named HB61 (5' TAATACGACT–CACTATAGG 3') and NEB OneTaq DNA Polymerase. 3.8µg of dsDNA was used as a template for *in vitro* transcription using NEB T7 polymerase and RNAseOUT inhibitor at 37°C overnight as per manufacturer's instructions. This yielded ~1000ng/µL of *cup-2* dsRNA and ~850ng/µL of *gfp* dsRNA that was used for RNAi injections after checking integrity of the RNA product by running it on a gel. One day past L4 XB737 worms were injected and their non-green progeny were analyzed by dissections and immunostaining.

### Feeding RNAi and whole mount DAPI

Feeding RNAi was performed as per standard procedures [132]. NGM plates supplemented with 100µg/mL Ampicillin and 1mM IPTG were prepared as per standard procedures [133]. The bacterial clones obtained from the Ahringer RNAi library were verified by sequencing and working stocks of verified clones were used for seeding RNAi plates. The bacteria were grown overnight in LB supplemented with 100µg/mL Ampicillin, seeded on to RNAi plates and allowed to grow for 2 days at room temperature. For each batch of RNAi experiments, a positive and negative control (empty vector) were included for comparison.

Synchronized L1s were obtained as above and were placed on plates at the desired temperature. Adult animals were analyzed 72 hours later by whole mount DAPI. Briefly, adult animals were washed off plates in PBS and fixed in methanol at -20°C overnight. Fixed animals were washed once in PBS then stained in 100ng/mL DAPI in PBS for 5 minutes, washed twice more, then mounted on 1% agarose pads for microscopy.

### Drug treatment assays

Worm preparation and analyses for DTT and Thapsigargin (TG) experiments were performed as for RNAi experiments described above. NGM plates containing the respective concentrations of drugs were made by adding drugs (or an equivalent amount of DMSO for no drug control plates) to the standard NGM agar recipe. DTT and Thapsigargin concentrations used were as previously described [88,89].

### Supporting information

**S1 Fig. A simplified schematic of the genetic pathway controlling the proliferation vs. differentiation balance in the *C. elegans* germline and the consequence of disruption of key**

**players in the pathway on the phenotype.** (A) Wild-type gonad. (B) *glp-1(gf)* gonad (C) *gld-2 (0) gld-1(0)* gonad. Proliferative cells in green, differentiating cells in red within the tube-like gonad.
(TIF)

**S2 Fig. Raw, average CUP-2 intensity measurements to compare CUP-2 levels in WT vs. *puf-8(q725)* genetic backgrounds at 25°C.** (A). CUP-2 immunostaining (by α-V5) in wild-type (WT) and *puf-8(q725)* dissected gonads. Both strains contain the *cup-2(ug1 [V5::2XFLAG::CUP-2])* allele. Scale bar = 10μm. (B) CUP-2 intensities measured by α-V5 immunostaining by drawing a line through the center of the germline from the distal end along the distal-proximal axis. Shaded area represents the standard deviation of average intensity measurements of each genotype. Fifteen germlines were analyzed for CUP-2 intensity measurements of each genotype. Arrowheads point to the average location of the transition zone measured in at least seven gonads of each genotype. Dashed line represents the predicted peak of PUF-8 expression, based on previous work that found that PUF-8's expression pattern is a bell-shaped curve centered around the transition zone with low expression levels in the distal end [54]. While we find that the overall CUP-2 expression levels are higher in *puf-8(q725)* gonads compared to wild type gonads, since this increase does not correlate with the known expression pattern of PUF-8 in wild type gonads, CUP-2 levels are unlikely to be directly regulated by PUF-8.
(TIF)

**S3 Fig. α-REC-8 and α-HIM-3 immunostaining of gonads of the indicated genotypes.** Asterisk, distal tip. Scale bar = 20μm
(TIF)

**S4 Fig. Raw, unscaled average SYGL-1 intensity measurements of individual experiments to compare SYGL-1 intensity in *glp-1(ar202)* genetic backgrounds.** SYGL-1 intensities were measured by α-FLAG immunostaining by drawing a line through the center of the germline from the distal end along the distal-proximal axis of the indicated genotypes. Each subfigure indicates an individual experiment comparing two genotypes that were processed together and imaged on the same slide with the same exposure setting. A-C measurements were used to generate scaled, fitted intensity curves shown in Fig 4B. Shaded area represents the standard deviation of average intensity measurements of each genotype. Fifteen germlines were analyzed for SYGL-1 intensity measurements of each genotype. Arrowheads point to the average location of the transition zone measured in at least seven gonads of each genotype. A. Average SYGL-1 intensity comparison of *sygl-1(am307)* against *cup-2(tm2838) sygl-1(am307)* germlines. B. Average SYGL-1 intensity comparison of *sygl-1(am307)* against *sygl-1(am307); glp-1 (ar202)* germlines. C. Average SYGL-1 intensity comparison of *sygl-1(am307)* against *cup-2 (tm2838) sygl-1(am307); glp-1(ar202)* germlines. D. Average SYGL-1 intensity comparison of *sygl-1(am307); glp-1(ar202)* against *cup-2(tm2838) sygl-1(am307); glp-1(ar202)* germlines
(TIF)

**S5 Fig. Effect of DTT treatment on weak Notch-dependent tumours and Notch-independent tumours.** A. Suppression of *puf-8(q725); glp-1(oz264)* tumours by DTT treatment. Phenotypes were analyzed by dissections followed by α-REC-8/α-HIM-3 staining. B. Quantification of phenotypic analysis of the effect of increasing doses of DTT on *glp-1(oz264)* and *glp-1(ar202)* tumours at 25°C. Phenotypes were analyzed by whole mount DAPI. C. Quantification of phenotypic analysis of the effect of increasing doses of DTT on Notch-independent *gld-2(q497) gld-1(q485)* tumours. Phenotypes were analyzed by dissections followed by α-

REC-8/α-HIM-3 staining.
(TIF)

**S6 Fig. Raw, unscaled average SYGL-1 intensity measurements of individual experiments to compare SYGL-1 intensity in wildtype and *glp-1(ar202)* genetic backgrounds with induction of ER stress by DTT treatment.** SYGL-1 intensities were measured by α-FLAG immunostaining by drawing a line through the center of the germline from the distal end along the distal-proximal axis of the indicated treatments. Each subfigure indicates an individual experiment comparing two treatments that were processed together and imaged on the same slide with the same exposure setting. A-B measurements were used to generate scaled, fitted intensity curves shown in Fig 8A, C-D measurements were used to generate scaled, fitted intensity curves shown in Fig 8B. Shaded area represents the standard deviation of average intensity measurements of each treatment. Fifteen germlines were analyzed for SYGL-1 intensity measurements of each treatment. Arrowheads point to the average location of the transition zone measured in at least seven gonads of each treatment. A. Average SYGL-1 intensity comparison of 0 DTT treated against 2 DTT treated *sygl-1(am307)* germlines. B. Average SYGL-1 intensity comparison of 0 DTT treated against 5 DTT treated *sygl-1(am307)* germlines. C. Average SYGL-1 intensity comparison of 0 DTT treated against 2 DTT treated *sygl-1 (am307); glp-1(ar202)* germlines. D. Average SYGL-1 intensity comparison of 0 DTT treated against 5 DTT treated *sygl-1(am307); glp-1(ar202)* germlines.
(TIF)

**S7 Fig. Raw, unscaled average SYGL-1 intensity measurements comparing *xbp-1* RNAi treated RNAi germlines against empty vector control RNAi treated germlines of *cup-2 (tm2838) sygl-1(am307); glp-1(ar202)*.** SYGL-1 intensities were measured by α-FLAG immunostaining by drawing a line through the center of the germline from the distal end along the distal-proximal axis of the indicated treatments. Shaded area represents the standard deviation of average intensity measurements of each treatment. At least twelve gonads were analyzed for SYGL-1 intensity measurements of each treatment.
(TIF)

**S1 Table. Key *glp-1* alleles used in this study.**
(DOCX)

**S2 Table. Phenotypic analysis of the effect of loss of *cup-2* on expression of the Glp phenotype.** Individual gonads were analyzed at 22.5°C by whole mount DAPI.
(DOCX)

**S3 Table. Phenotypic analysis of the effect of ER stress induced by DTT treatment on suppression of Notch-dependent tumourous phenotypes.** *puf-8(q725); glp-1(oz264)* was analyzed by dissections followed by α-REC-8/α-HIM-3 staining while the *glp-1(oz264)* and *glp-1 (ar202)* phenotypes was analyzed by whole mount DAPI.
(DOCX)

**S4 Table. Phenotypic analysis of the effect of ER stress induced by DTT treatment on suppression of a *gld-2 gld-1* Notch-independent tumours.** Phenotypes were analyzed by dissections followed by α-REC-8/α-HIM-3 staining.
(DOCX)

**S5 Table. List of strains used.**
(DOCX)

## Acknowledgments

The original strain containing the *glp-1(ar224)* allele was a gift from Jane Hubbard. We thank Tim Schedl for the strain containing *sygl-1(am307)*. Strains FX02838 *cup-2(tm2838)* and FX06098 *der-2(tm6098)* were provided by the National BioResource Project (NBRP), Japan. Some strains were provided by the CGC, which is funded by NIH Office of Research Infrastructure Programs (P40 OD010440). The HIM-3 antibody was a gift from Monique Zetka. The REC-8 antibody was a gift from Pavel Pasierbek and Joseph Loidl. We thank Marcus Samuel for use of the confocal microscope. We thank members of the Hansen lab, Paul Mains, Jim McGhee and Tim Schedl for helpful discussions.

## Author Contributions

**Conceptualization:** Ramya Singh, Ryan B. Smit, Dave Hansen.

**Data curation:** Ramya Singh.

**Funding acquisition:** Dave Hansen.

**Investigation:** Ramya Singh, Ryan B. Smit, Xin Wang, Chris Wang, Hilary Racher.

**Resources:** Dave Hansen.

**Supervision:** Dave Hansen.

**Visualization:** Ramya Singh.

**Writing – original draft:** Ramya Singh, Ryan B. Smit, Dave Hansen.

**Writing – review & editing:** Ramya Singh, Ryan B. Smit, Dave Hansen.

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
