## [Decision Letter · Decision Letter 0]

3 Aug 2021

Dear David,

Thank you very much for submitting your Research Article entitled 'Reduction of Derlin activity suppresses Notch-dependent tumours in the C. elegans germ line' to PLOS Genetics.

The manuscript was fully evaluated at the editorial level and by independent peer reviewers. The reviewers appreciated the attention to an important topic but identified some minor concerns that we ask you address in a revised manuscript.

In particular, please address the reviewers' comments/suggestions regarding minor changes needed in the text and figures. The experiment suggested by Reviewer #3 would further strengthen this study and support your model, but is not required for publication of this manuscript, so whether this is included in the revised manuscript will be left at your discretion.

[LINK]

Yours sincerely,

Mónica P. Colaiácovo

Associate Editor

PLOS Genetics

Gregory P. Copenhaver

Editor-in-Chief

PLOS Genetics

Reviewer's Responses to Questions

**Comments to the Authors:**

Reviewer #1: Comments:

The C. elegans germline provides a premier experimental system for studying the cell and developmental biology of stem cells. In this system, the somatic distal tip cell produces Delta-like ligands to activate the GLP-1/Notch receptor in the germline thereby maintaining a pool of germline stem cells in the niche. In the absence of GLP-1/Notch signaling, stem cells are not maintained causing all germ cells to differentiate prematurely prior to proliferation, which causes sterility. By contrast, inappropriate activation of GLP-1/Notch signaling results in the over-proliferation of germ cells and germline tumor formation. The Notch signaling pathway, besides representing one of the major intercellular signaling pathways needed for metazoan development, is of importance from the standpoint of human cancer biology. Activating mutations in the NOTCH1 receptor are found in a large percentage of human T cell acute lymphoblastic leukemia cases. Thus, understanding how Notch signaling output might be modulated in vivo has therapeutic ramifications.

The present study extends important work from the Hansen lab on GLP-1/Notch signaling. Here they present the novel finding that Derlin, a key protein factor in the endoplasmic reticulum (ER)-associated protein degradation (ERAD) pathway promotes GLP-1/Notch signaling. Notably, they show that elimination of Derlin function can suppress germline tumor formation by constitutively activated forms of the GLP-1/Notch receptor. Remarkably, their data attribute this suppression to the activation of the unfolded protein response (UPR) in the absence of a fully functional ERAD response. The idea here is that when the ERAD pathway is defective, misfolded proteins accumulate in the ER, which activates the UPR and concomitantly reduces the signaling output from constitutively activated GLP-1/Notch receptors. Though the mechanism by which UPR activation is protective remains to be determined, the authors discuss several reasonable and testable hypotheses in their Discussion. Like many great studies, this work raises many important questions for future work, which will undoubtedly be pursued by investigators studying multiple stem cell systems. The authors should consider addressing the following specific points.

Specific Points

1. Page 5, first paragraph. Consider citing a review about genetic redundancy and robustness of developmental signaling pathways.

2. Page 6, first paragraph. Consider citing a more recent review of ERAD. There is quite a nice one in the June 2021 issue of Molecular Cell.

3. Page 7, second paragraph. Consider adding an introductory figure to the main text or supplement. This will help general readers understand the experiments to simultaneously disrupt gld-1 and gld-2 function.

4. Page 8, last paragraph. This section may be difficult for general readers to follow without some additional explanation. Alternatively, PUF-8, it’s molecular identity, and the prior findings (including those to be published elsewhere), could explained at the start of the Results section. My sense is that this portion of the Introduction, in its present form, not only disrupts the flow of the explication of the study’s chief findings, but also will lose general readers. Perhaps the Introduction should focus on what was found and the surrounding context and not exactly how it was found, which can be explained in the Results section.

5. Page 10. Results section. The authors utilize a variety of alleles in their genetic analyses. Many of these are well known to the cognoscenti in the field. However, general readers and newcomers to the field might find the presentation difficult to understand. Consider putting a table in the supplement summarizing the nature of key alleles used in the genetic analyses and appropriate references.

6. Page 10, last paragraph and Figure 1B. Please apply statistical analyses to the data shown in Figure 1B.

7. Page 10, first paragraph and Figures 2B and 2C. Consider reorganizing Figure 2B so the results you discuss first come first. Also cite Figures 2B and 2C as appropriate.

8. Page 17, and Figure 4A. Ideally, Figure 4A should contain the wild type and cup-2 single mutants. On my computer screen, the anti-FLAG staining in Figure 4A actually looks extended in the tm2838; ar202 double.

9. Page 18, second paragraph. Please refer to “unpublished data” as opposed to “data not shown.”

10. Page 23, second paragraph. Is cup-2 synthetically lethal with xbp-1? If not, then the suggestion is to use an xbp-1 null allele (e.g., zc12) to complement the RNAi experiment—this is an important conclusion and an additional genetic test seems warranted.

11. Page 24, second paragraph. You mean distal shift, right?

12. Page 28, first paragraph. Is it also worthwhile considering the levels or activities of the membrane-localized enzymes that process the GLP-1/Notch receptor?

13. Page 31, first paragraph. The extracellular portion of the Notch receptor is luminal in the ER. The reviews I've read point out mechanistic differences in ERAD responses triggered by misfolded domains in the lumen, membrane, and cytoplasm. Might be worth a mention.

14. I am not sure this is worth a mention in the Discussion, but it is perhaps worth thinking about. Previous studies from the Hansen, Schedl, and a few other labs have shown that various perturbations of the spliceosome can promote germline tumor formation. In the landmark paper on IRE1-dependent splicing of HAC1 mRNA, it was shown that defects in spliceosome-mediated splicing by themselves induce the UPR in budding yeast (Sidrauski et al., 1996). One would guess this happens in the worm, which would be a condition that would, according to the current manuscript, suppress not enhance tumor formation. No doubt that the perturbation of splicing affects multiple steps in the pathway (e.g., the gld-1 pathway) may provide some explanation. Yet, these considerations might mean that targeting ERAD or the UPR for therapeutics might be complicated and even counterproductive in some situations.

David Greenstein

Reviewer #2: The review is uploaded as an attachment.

Reviewer #3: In this manuscript, Dr. Hansen and colleagues present a detailed study of Derlin knockout-mediated suppression of germline tumor phenotype resulting from gain-of-function mutations in Notch receptor glp-1. This suppression was observed mainly for one of two nematode Derlin homologs, cup-2, with weaker contribution from the other homolog, der-2. Loss of cup-2 was associated with a reduced accumulation of GLP-1/Notch transcriptional target SYGL-1. The authors document expression of CUP-2 in germline cells, and its localization to the ER. The authors provide strong evidence that knockout of Derlins activates ER stress and the Unfolded Protein Response which reduce aberrant Notch signaling. This makes a novel contribution to our understanding of general cellular mechanisms affecting the strength of Notch signaling in the context of tumorigenic gain-of-function mutations

The study is thorough and the conclusions are well-supported by the data.

To strengthen the overall conclusions of the study, especially in the context of the last paragraph of the Discussion, the authors could distinguish whether suppression of tumor acts on any level Notch signaling or only on the Notch gain-of-function alleles.

The genetic suppression of Notch-dependent tumors by cup-2 loss could be explained if Derlins buffered the germlines expressing temperature-sensitive glp-1 mutants from ER stress so that these genotypes were “addicted” to Derlins and sensitized to the loss of Derlin activity. This is consistent with cup-2(0) leading to a stronger reduction of Notch signaling mediated by glp-1(gf) mutants compared to the otherwise wt background.

If this were the case, loss of cup-2 would not suppress a glp-1-dependent tumor where glp-1 is wild type, so not in need of ER stress buffer, such as gld-2; teg-1 tumor (Wang et al., 2012) or gld-2 prp-17 tumor (Kerins et al., 2010). Additionally, such data would provide a stronger support to the proposal that drugs affecting Derlin activity would predominantly target tissues with aberrant rather than normal Notch signaling.

The authors should also address the following points:

The title of Fig. 6 needs to be revised, the experiment in Fig. 6B does not support the conclusion that “cup-2 activity partially requires retrograde transport”, as partial suppression of tumor by sorting nexin knockout doesn’t speak to whether it happens through the same or different mechanism than cup-2-mediated suppression. Simply stating that “Retrograde transport might contribute to cup-2 activity” as done in the Results would be more accurate.

Additionally, the legends for panels A and B are switched, which should be corrected.

In Fig. 8B, the graph for 0mM DTT treatment doesn’t have standard deviation.

I suggest modifying the color scheme of the graphs representing tumor phenotype analysis: in Figs. 1 and 6 green represents wild type germlines, while in Figs. 3, 7, and S3 green represents tumorous germlines. It would help if one color scheme were used consistently.

In the last paragraph of the Results, the authors state that “increasing doses of DTT lead to a more proximal shift of the peak of SYGL-1 expression”… This is confusing because the Figure shows SYGL-1 shift distally (closer to the distal tip of the germline). Perhaps distal/proximal is this paragraph could be restated so that proximity to the tip of the germline is not in conflict with anatomical definitions of distal/proximal ends of germline.

**Have all data underlying the figures and results presented in the manuscript been provided?**

Reviewer #1: Yes

Reviewer #2: Yes

Reviewer #3: Yes

PLOS authors have the option to publish the peer review history of their article (what does this mean?). If published, this will include your full peer review and any attached files.

Reviewer #1: **Yes: **David Greenstein

Reviewer #2: No

Reviewer #3: No

---

## [Editor Report · Decision Letter 1]

8 Sep 2021

Dear Dr Hansen,

Thank you for carefully addressing the reviewers' comments. We are pleased to inform you that your manuscript entitled "Reduction of Derlin activity suppresses Notch-dependent tumours in the C. elegans germ line" has been editorially accepted for publication in PLOS Genetics. Congratulations!

Yours sincerely,

Mónica P. Colaiácovo

Associate Editor

PLOS Genetics

Gregory P. Copenhaver

Editor-in-Chief

PLOS Genetics

Comments from the reviewers (if applicable):

**Data Deposition**

http://datadryad.org/submit?journalID=pgenetics&manu=PGENETICS-D-21-00876R1

**Press Queries**

---

## [Editor Report · Acceptance letter]

17 Sep 2021

PGENETICS-D-21-00876R1 

Reduction of Derlin activity suppresses Notch-dependent tumours in the C. elegans germ line 

Dear Dr Hansen, 

We are pleased to inform you that your manuscript entitled "Reduction of Derlin activity suppresses Notch-dependent tumours in the C. elegans germ line" has been formally accepted for publication in PLOS Genetics! Your manuscript is now with our production department and you will be notified of the publication date in due course.

With kind regards,

Katalin Szabo

PLOS Genetics

On behalf of:
